# Obstacle Avoidance and Path Planning Methods for Autonomous Navigation of Mobile Robot

**DOI:** 10.3390/s24113573

**Published:** 2024-06-01

**Authors:** Kornél Katona, Husam A. Neamah, Péter Korondi

**Affiliations:** Department of Electrical Engineering and Mechatronics, Faculty of Engineering, University of Debrecen, 4028 Debrecen, Hungary; husam@eng.unideb.hu (H.A.N.); korondi.peter@eng.unideb.hu (P.K.)

**Keywords:** obstacle avoidance, global path planning, local path planning, autonomous vehicles, navigation algorithms

## Abstract

Path planning creates the shortest path from the source to the destination based on sensory information obtained from the environment. Within path planning, obstacle avoidance is a crucial task in robotics, as the autonomous operation of robots needs to reach their destination without collisions. Obstacle avoidance algorithms play a key role in robotics and autonomous vehicles. These algorithms enable robots to navigate their environment efficiently, minimizing the risk of collisions and safely avoiding obstacles. This article provides an overview of key obstacle avoidance algorithms, including classic techniques such as the Bug algorithm and Dijkstra’s algorithm, and newer developments like genetic algorithms and approaches based on neural networks. It analyzes in detail the advantages, limitations, and application areas of these algorithms and highlights current research directions in obstacle avoidance robotics. This article aims to provide comprehensive insight into the current state and prospects of obstacle avoidance algorithms in robotics applications. It also mentions the use of predictive methods and deep learning strategies.

## 1. Introduction

Autonomous robots are machines or devices capable of operating independently and making decisions in their environment. These robots are equipped with sensors and embedded systems to gather information about their surroundings, such as mapping, navigation, and obstacle detection. Obstacle avoidance plays a crucial role in the operation of autonomous robots, enabling them to navigate their environment efficiently and safely. Obstacle avoidance algorithms assist robots in avoiding obstacles and minimizing collisions, allowing them to reach their destination safely and accomplish their tasks. Thus, obstacle avoidance is an indispensable element of effective and reliable operation for autonomous robots. This paper explores a literature review of alternative route planning and mobile robot navigation methods. The main algorithms it considers are discussed in the following sections. Global path planning involves navigating a robot based on preexisting environmental data, which is loaded into the robot’s planning system to compute a trajectory from the starting point to the destination. This method generates a complete path before the robot begins its journey, essentially optimizing the route gradually [1]. Global path planning is consciously determining the best way to move a robot from a starting point to a destination. In global route planning, the robot has already been moved from the starting location to the destination, and the robot is then released into the specified environment [2]. In contrast, local path planning involves navigating a robot in dynamic or unknown environments where the algorithm adapts to real-time obstacles and changes. This method primarily focuses on real-time obstacle avoidance using sensor-based data for safe navigation [3]. The robot typically follows the shortest, straight-line path from the start to the destination until encountering an obstacle. Upon detection, it deviates from this path while updating essential details like the new distance to the target and the point of obstacle bypass [2]. Continuous knowledge of the target’s position relative to the robot is critical for accurate navigation, as depicted in Figure 1.

The diagram of the algorithms of each type included in this paper is shown in Figure 2. Another classification divides the methods into classical and heuristic algorithms (Figure 3). The classification according to classic and heuristic obstacle avoidance algorithms makes the selection and application of algorithms more transparent and manageable. Users can more easily identify which algorithm best meets the requirements of a given problem. Classical algorithms such as Dijkstra perform well for minor deterministic problems, while heuristic algorithms such as A* can be more efficient for larger and more complex issues. Moreover, the separation between global and local search algorithms is less clear. There are heuristic algorithms (such as A* or the DL-based algorithms) that have both versions of the search algorithm. This paper follows the latter classification.

One group of algorithms is called optimization methods. These mathematical procedures and algorithms aim to find the best possible solution to a given problem within the constraints available. An optimal solution is usually a combination of the values of one or more variables that maximizes or minimizes the value of the objective function while taking into account various constraints or conditions. For example, Particle Swarm Optimization, Cuckoo Search Algorithm, Artificial Bee Colony, Ant Colony Optimization, and Grey Wolf Optimization are based on such optimization techniques. These methods are called swarm (population)-based because they are inspired by animal behavior. Usually, some population of individuals (solutions) is used, and these individuals are iteratively developed and modified to find the best solution. They can effectively find optimal solutions to complex and diverse problems that traditional algorithms cannot manage with difficulty or at all.

Section 2 discusses bright spaces and fundamental obstacle avoidance methods. Then, in the Section 3, classical avoidance algorithms such as Dijkstra, Floyd–Warshall (FW), Bellman-Ford (BF), Artificial Potential Field (APF), Bug Algorithms, Vector Field Histogram (VFH), Probabilistic Roadmap Method (PRM), Rapidly exploring Random Tree (RRT), Cell Decomposition (CD), and the Following Gap Method (FGM) are discussed. Heuristic algorithms are presented in Section 3. This includes the A* Algorithm, Fuzzy Logic (FL), Particle Swarm Optimization (PSO), Genetic Algorithm (GA), Cuckoo Search Algorithm (CSA), Artificial Bee Colony (ABC), and Ant Colony Optimization (ACO). It also mentions using deep learning (DL) strategies and predictive methods. In this section, we discuss Artificial Neural Networks (ANNs), Model Predictive Control (MPC), and Deep Reinforcement Learning (DRL). Other algorithms are mentioned here, such as Dynamic Window Approach (DWA), Golden Jackal Optimization (GJO), or Grey Wolf Optimization (GWO). At the end of the chapter, a further so-called hybrid algorithm consists of two or more of the algorithms discussed earlier. Sliding Mode (SM) is presented in more detail among the hybrid methods. This article tries to collect and analyze most of the algorithms commonly used in practice. However, covering all currently existing methods in a single article is impossible, so this is not the aim here. This paper attempts to provide an overview of historically important and currently significant algorithms in practice as comprehensively as possible. Of course, it is impossible to discuss all possible methods (especially in the case of hybrid algorithms), but we tried to present the development directions of each algorithm. Such an extensive literature review cannot be found in other works. One of this article’s most important values, after the description of the theoretical background, is the summary table of the individual algorithms, which provides a sufficient comparison based on the algorithms’ main properties (e.g., convergence, calculation time).

## 2. Classic Approaches

This section presents some classic approaches.

### 2.1. Dijkstra Algorithm

Dutch scientist Edsger Wybe Dijkstra introduced the Dijkstra Algorithm (DA) in 1956, which he published in 1959 [5]. The question of the shortest path between two nodes in a directed graph is solved by this method, which is one of the most commonly used techniques for mapping isolated workspace paths [6]. This method is a well-known strategy, but it is less effective when the origin and destination are farther apart. In this case, the algorithm calculates the shortest path for all nodes, even if the node is irrelevant for the optimal route. Consequently, most of the calculations may be redundant, resulting in a time-consuming process. Another factor that may contribute to the time-consuming process is the presence of long edges in the graph. In this case, the Dijkstra algorithm has to spend a considerable amount of time processing the edges [7].

To plan the shortest path in Dijkstra’s algorithm, the starting position must be specified, and two heaps *S* and *U* must be introduced. The S heap records the vertices for which the shortest path is not found and the distance between the vertex and the starting point [8]. The flowchart of the Dijkstra algorithm is shown in Figure 4.

The work in [9] used DA to define vehicle routes on toll roads. Path planning is in a localization-insecure environment based on the Dijkstra method in [10]. Dijkstra was used to determine the shortest distance between cities on the island of Java [11]. This method was later modified to handle the situation where most of the network parameters are unknown and expressed as neutrosophic values (can be true, false, and neutral simultaneously, depending on the point of view) [12]. A Dijkstra-based route planning strategy for autonomous vehicles is included in [13]. In [14], a Dijkstra algorithm is applied to unmanned aerial vehicles (UAVs). Ref. [15] presents the optimal route planning of an unmanned surface vehicle in a real-time maritime environment using the Dijkstra algorithm.

### 2.2. Floyd-Warshall Algorithm (FW)

The Floyd-Warshall algorithm can be considered dynamic programming, and it was published in 1962 by Robert Floyd. The algorithm efficiently and simultaneously finds the shortest paths between all pairs of vertices of a weighted and potentially directed graph [16,17].

The algorithm compares all possible paths for each line of all points on the graph. The graph’s vertices should be numbered from 1 to *n* (*n* number of vertexes). Suppose there is also a shortest path function f(i,j,k) which gives the shortest path from *i* to *j* using only the node from 1 to *k* as an intermediate point. The ultimate goal of using this function is to find the shortest path from each vertex *i* to vertex *j* using the intermediate node from 1 to k+1. This algorithm first computes the function f(i,j,1) for each pair (i,j), then uses the results to compute f(i,j,2) for each pair (i,j), and so on. This process continues until k=n, and the shortest path is found for all (i,j) pairs with the interpolation of vertices [18].

The algorithm consists of two parts: the construction of the path matrix and the state transition equation. The construction of the path matrix is based on the weight matrix of the graph to obtain the matrix $D^{n}$ of the shortest path between each two points. The elements of row *i* and column *j* of the matrix $D^{n}$ are the length of the shortest path from vertex *i* to vertex *j*. The state transition equation mathematically calculates the shortest distance between each point (Equation 1). The computation time is $O(n^{3})$ [19].
(1)$$d_{ij}^{k}=min\left\{d_{ij}^{k-1},d_{ik}^{k-1}+d_{kj}^{k-1}\right\}$$
where the notation $d_{ij}^{k}$ represents the shortest path from *i* to *j* that also passes through vertex *k*. For example, $d_{ij}^{0}$ is the edge length between vertices *i* and *j*.

### 2.3. Bellman-Ford Algorithm (BF)

The Bellman-Ford algorithm is a classical method that computes the shortest paths in a weighted graph from a single source. This algorithm considers the negative-weighted edges of the graph, so it can handle graphs that contain negative-weighted cycles. These cycles generate several paths from the origin to the destination, where each cycle minimizes the shortest path length. The algorithm efficiently uses $O(n\dot{m})$ time for a graph with *n* vertices and *m* edges. The BF algorithm can handle edges with negative weights, unlike Dijkstra’s algorithm, which only works with edges with positive weights. For this reason, the BF algorithm is mainly used for graphs with negative edge weights. Although its efficiency is lower than that of Dijkstra’s algorithm, some problems would be impossible without negative weights. The BF algorithm is similar to Dijkstra’s algorithm, but it approximates all edges instead of selecting vertices with minimum distance. This operation is performed n−1 times, where *n* is the number of vertices in the graph, and these iterations provide an exact prior in the graph [20,21,22].

### 2.4. Artificial Potential Field (APF)

The idea is that the mobile robot moves within a potential field where the robot and obstacles behave as positive charges while the target behaves as a negative charge. The mismatch between attractive and repulsive forces helps the robot to move in the environment. The attractive force attracts the robot to the target location, while the repulsive force keeps it away from each obstacle [23], as shown in Figure 5.

The final force acting on the robot is the vector sum of all repulsive and attractive forces. However, the distance determines the magnitude of the force, i.e., obstacles close to the robot will have a more significant effect. Similarly, if the robot is far from the target, its speed will be high and slow down as it approaches the target. As mentioned in the literature [24], the attractive force is the negative gradient of the attractive potential (Equation 2).
(2)$$F_{attr}=-\nabla U_{attr}=-K_{attr}\left(d-d_{goal}\right)$$
where $d-d_{goal}$ is the Euclidean distance between the current position and the target, and $K_{attr}$ is the scaling factor. The repulsive force can be calculated by adding the repulsive effect of the obstacles on the robot. This can be obtained by calculating the obstacles’ distance and direction (angle) from the robot. An obstacle close to the robot has a high repulsive force. The formula described by [25] is (Equation 3)
(3)$$U_{rep}=\sum_{i=1}^{n}U_{repi}\left(d\right)$$

$U_{rep}$ negative gradient repulsive force. So (Equation 4),
(4)$$F_{rep}=-U_{repi}\left(d\right)$$

To avoid local minima, various methods have been devised. One such method is the left-turning potential field approach, which compels the robot to change direction when encountering a local minimum. Conversely, the virtual target point method involves strategically placing a virtual target point when the robot reaches a local minimum. During this process, the robot disregards the influence of both the target point and obstacles, enabling it to pivot and break free from the local minimum. Another critical issue with APF is its susceptibility to local minima, which can hinder the robot’s progress. Symmetric and U-shaped obstacles exemplify these dead-end scenarios, leading to the robot becoming trapped. Figure 6 illustrates symmetric obstacles, where the forces exerted by the target and obstacles cancel each other out, resulting in a stalemate for the robot—a classic instance of local minima. To address this problem, significant attractor forces are temporarily applied at random locations to prevent the robot from being trapped in local minima. These measures aim to disrupt the equilibrium between attractive and repulsive forces, enabling the robot to navigate effectively [26]. In summary, while APF offers a direct path from source to destination, its susceptibility to local minima poses a significant challenge. Various strategies, such as the left-turning potential field and virtual target point methods, have been developed to mitigate this issue and ensure smoother navigation in complex environments [27].

The APF has been used in a dynamically changing, obstacle-filled environment between unmanned aerial vehicles (UAVs) [28]. Ref. [29] proposes an improved artificial potential field method for autonomous underwater vehicle (AUV) route planning. Based on an improved artificial potential field, ref. [30] introduces dynamic route planning for autonomous vehicles on icy and snowy roads. Ref. [31] discusses local path planning for multi-robot systems using an improved APF. Furthermore, ref. [32] outlines an active obstacle avoidance method for autonomous vehicles that is also based on an improved APF.

### 2.5. Bug Algorithms

Despite the presence of more efficient algorithms, Bug algorithms are still significant in robotics. These were the earliest navigation and obstacle avoidance algorithms that achieved relatively reliable results with speedy computation times. The algorithms are designed to work assuming that the robot is a single point in 2D space and that its movement is between each point. Bug algorithms are a popular type of robot navigation algorithms that provide a trajectory following an obstacle boundary in navigation scenarios with unknown obstacles, similar to the behavior of a bug [33]. The algorithm can be divided into three main variants based on their obstacle avoidance behavior, as discussed below [34,35]:The Bug-1 algorithm activates when the robot detects an obstacle. It starts circumnavigating the obstacle until it reaches the starting point from which it began while calculating the shortest distance from the destination to the departure point and creating a new path from the calculated departure point to the destination as it circumnavigates the obstacle. After completing full circles, it resumes circumnavigating the obstacles until it reaches the departure point, then proceeds on the newly generated path toward the destination.The bug-2 algorithm sets a direction from the starting position to the destination, and the robot follows it until it encounters an obstacle. Upon interruption, it follows the obstacle’s edge and calculates a new direction from each new position until the new direction matches the original direction. Once reaching this position, the robot resumes following the previously generated path towards the destination.In contrast, the Dist-Bug algorithm relies on distances to targets and obstacles. When encountering an obstacle on the path, the robot begins following the obstacle’s edge and calculates the distance between that point and the destination at each point. The point with the smallest distance to the target is called the distance point. Subsequently, the robot creates a new path along which it moves to the destination when it finds the distance point during its movement around the obstacle.

The three versions are shown in Figure 7.

A maritime search route planning method for unmanned surface vehicles (USVs) based on the improved Bug algorithm presented here is also presented in [36].

### 2.6. Follow the Gap Method (FGM)

The FGM avoids obstacles by finding the gap between them. It calculates the gap angle. The minimum gap between obstacles is the threshold gap from which the robot can move. If the measured gap is larger than the threshold, the robot follows the calculated gap angle. Obstacle avoidance using the FGM is achieved in three main steps [37].

The algorithm uses sensory information to identify gaps with the largest angle and works in three steps, as follows [38]:The initial step involves computing the arrays of gaps. During this phase, the algorithm utilizes the current sensory data, such as information from the LIDAR sensor, to produce a gap array. This array provides details regarding the sizes of the available gaps surrounding the robot in angular form. The FGM algorithm identifies the largest gap by the conclusion of this stage.The FGM calculates the angle to the gap’s center point using specific geometric relations.In the third stage, this method calculates the final heading angle, $\phi_{final}$, using (Equation 5)
(5)$$\phi_{final}=\frac{\frac{\alpha }{d_{min}}\phi_{gap-c}+\phi_{goal}}{\frac{\alpha }{d_{min}}+1}$$The weighted function described in Equation (Equation 5) comprises the angle to the center point of the widest gap $\phi_{gap-c}$, the angle to the goal point $\phi_{goal}$, the distance to the nearest obstacle $d_{min}$, and a safety parameter denoted as α. Higher values of the alpha parameter prompt the robot to maintain a safe distance from obstacles and align with the center of the safe gap. Conversely, lower values of alpha lead the robot to prioritize the goal point, potentially approaching obstacles too closely in certain scenarios.

The representation of the gaps accessible to the robot, the angle towards the midpoint of the widest gap, the angle towards the destination, and the final heading angle determined by FGM are depicted in a robot-obstacle configuration, as illustrated in Figure 8.

So, in this procedure, the robot selects the largest gap around it and moves towards the target, taking into account the largest gap and the minimum distance from the obstacle. One of the drawbacks of this method is the lengthening of the path, which can sometimes be unnecessary. Another challenge is the subtle differences in gap sizes. This can sometimes result in the robot changing the number of selected gaps instantly, which can lead to zigzag paths [39].

In [40], the collision avoidance task is accomplished with the Follow the Gap Vector Method. A central part of the approach proposed in [2] is to identify gaps in the environment by analyzing sensor data.

### 2.7. Vector Field Histogram (VFH)

The algorithm initiates by generating a 2D histogram around the robot to depict obstacles. Subsequently, the 2D histogram undergoes updates with new sensor detections. It converts this 2D histogram into a 1D histogram and further into a polar histogram. Finally, the algorithm identifies the most suitable sector characterized by low polar obstacle density and computes the steering angle and velocity towards this direction. Figure 9 is from the work of [41] which illustrates the 2D histogram grid. The conversion from 2D to 1D histogram is shown in Figure 10a, and Figure 10b is a representation of the 1D polar histogram with obstacle density for a situation where the robot has three obstacles, *A*, *B* and *C* in its close vicinity.

The first step is to sort the costs of the traversable area and then calculate the cost using the cost function based on the indicated direction of the polar histogram. The designated directions are selected from the traversable areas taking into account the robot’s kinematic and dynamic characteristics. Inaccessible sectors, as determined by the robot’s capabilities, are classified as impassable areas. Areas above the threshold are labeled as impassable, whereas those below the threshold are considered passable. To continue, the histogram generated in the previous step must be converted into a binary format by choosing the appropriate threshold based on the current situation. The commonly used cost function is shown as follows (Equation 6) [43]:(6)$$f\left(v\right)=c_{1}\cdot \Delta \left(v,d_{g}\right)+c_{2}\cdot \Delta \left(v,\frac{\Theta }{\alpha }\right)+c_{3}+\cdot \Delta \left(v,d_{g-1}\right)$$

The candidate direction fv represents the cost value f(v). $c_{1}$, $c_{2}$, and $c_{3}$ are three parameters to be determined according to the actual situation. The $d_{g}$ is the target direction, $d_{g-1}$ is the previous direction, and the orientation of the robot is $\frac{\Theta }{\alpha }$. The absolute difference between *v* and $d_{g}$ is denoted by $\Delta (v,d_{t})$. The difference between the marked direction and the orientation of the robot is denoted by $\Delta \left(v,\frac{Theta}{alpha}\right)$. The difference between *v* and $d_{g-1}$ is denoted by $\Delta (v,d_{g-1})$.

To determine the robot’s desired control command, this algorithm employs a two-stage data reduction process. Although this ensures accurate computation of the robot’s path to the target, it necessitates additional resources, such as memory and processing power [44].

Initial tests have shown that the mobile robot can use the VHF to traverse very crowded obstacle courses at high average speeds, and can pass through narrow openings (e.g., doorways) and move through narrow corridors without oscillating [41]. In [45], an improved 3D-VFH algorithm is proposed for autonomous flight and local obstacle avoidance of multirotor UAVs in confined environments.

### 2.8. Cell Decomposition (CD)

The cell-by-cell technique divides the area into non-overlapping grids, called cells, and uses grids that can be connected from the initial cells to the target to move from one cell to another. This method is classified as exact, approximate, and probabilistic CD depending on the assignment of boundaries between cells. For exact CD, the resolution is lossless and the shape and size of the cells are not fixed and each element is assigned a number. In contrast, for approximate CD, the decomposition result approximates the actual map and the grid has a fixed shape and size. And the probabilistic CD is like the approximate CD, except for the cell boundaries, which do not represent a physical meaning [46]. Figure 11 shows that the CD systems can be divided into three classes.

### 2.9. Probabilistic Roadmap Method (PRM)

Classical methods face several drawbacks, such as high time requirements at large scales and getting bogged down in local minima, which makes them ineffective in practical scenarios. To address these limitations and increase efficiency, probabilistic algorithms have been proposed. These algorithms aim at providing practical paths for robots through static workspaces [48].

One of the most important examples is the Probabilistic Roadmap Method (PRM) [49]. It uses lines to delimit the connectivity of the robot’s free areas. This includes the visibility graph and the Voronoi graph [47]. Figure 12 illustrates these two graphs.

In the visibility graph, the obstacles are represented as polygons [50], and the vertical nodes of the polygonal obstacles are connected in such a way that the path length is minimized while the lines remain close to the obstacles. In contrast, the Voronoi graph uses the two closest points of the edges of the obstacles for planning and divides the domain into subdomains. In the latter case, the robot moves farther away from the obstacles, which increases safety but results in longer paths compared with the visibility graph [51].

To link the initial state with the goal region, PRMs explore this roadmap graph and pinpoint a sequence of states and local connections that the robot can traverse. While these algorithms can theoretically create arbitrarily accurate representations as the number of samples approaches infinity, in practice, only a handful of critical states are needed to define solution trajectories. These critical states often have significant structure, such as entries to narrow passages, but they can only be identified through exhaustive sampling [52].

These algorithms offer highly accurate representations with a theoretically infinite number of samples. In practice, however, this is only necessary in a few cases. For example, entering a bottleneck. The Voronoi graph continues to play a crucial role in the further development of different algorithms for different purposes [53]. Notably, ref. [54] presents a useful visibility Voronoi graph search algorithm for generating routes for unmanned surface vehicles. In addition, ref. [55] uses the Voronoi graph to partition agricultural areas into multiple fields, making it easier for multiple robots to perform agricultural tasks.

### 2.10. Rapidly Exploring Random Tree (RRT)

The Rapidly exploring Random Tree (RRT) method facilitates swift exploration of the configuration space [56]. Initially proposed by LaValle [57], the RRT algorithm generates a graph, termed a “tree”, where nodes signify potential reachable states and edges denote transitions between states. The RRT’s root denotes the initial state, with all other states reachable along the path from the root to the corresponding node. Leveraging a sampling approach, this algorithm operates effectively in complex environments, evading local minima [58]. It has proven effective in tackling nonholonomic and kinodynamic motion planning challenges. In robotics, algorithms employed to generate RRTs are versatile, allowing trajectories to incorporate turns at any angle, albeit subject to kinematic and dynamic constraints [56].

When sampling, it allows all nodes in the robot configuration space to be reached with equal probability. Based on the constraints of the algorithm, it selects a node in the random tree. On impact, it resamples and discards the previous node. If no collision occurs, the selected node is added to the random tree. If a node on the route is redundant, it is deleted; otherwise, it remains as a node in the random tree [59].

The flow chart of an RRT is shown in Figure 13:

This method does not require the modeling of space and can be used in large-scale environments. It also takes into account the objective constraints of unmanned vehicles, making it suitable for handling route planning problems in dynamic and multiobstacle environments. However, the route is randomly generated, leading to distortion. Second, the random tree has no orientation during the search process, resulting in slow convergence speed and low search efficiency [60]. Several improvements have been made to address the limitations of the algorithm. Among others, the RRT-Connect algorithm, the asymptotically optimal Rapidly Exploring Random Tree (RRT), the asymptotically optimal bidirectional Rapidly Exploring Random Tree (B-RRT), and the intelligent bidirectional RRT (IB-RRT) were born out of this necessity. Ref. [60]. The RRT* algorithm can construct an RRT whose branches converge asymptotically to the optimal solution given a given cost function. It solves feasibility problems efficiently and qualitatively concerning the cost function [56].

RRT has been used to plan the routes of ships [58], industrial robots [59] and micro aerial vehicles (MAVs) [61], among others.

## 3. Heuristic Approach

A heuristic approach is used to solve problems faster [62]. The method has proven its effectiveness and is widely used in autonomous navigation [5].

### 3.1. A* Algorithm

The A* algorithm is a graph search algorithm similar to Dijkstra’s algorithm, developed by Hart (1968) [63] to speed up the search process of Dijkstra’s algorithm. To do this, they introduced a heuristic cost function, which is the distance between the current point and the target point. Like the Dijkstra algorithm, the A* algorithm needs an environment model, e.g., a grid map. In the A* algorithm, the search area is usually divided into small squares, where each square represents a node. The algorithm can solve various routing problems with superior performance and accuracy compared with Dijkstra’s algorithm. Algorithm A* solves problems by finding the path with the lowest cost (e.g., the shortest time) among all possible paths to the solution. Of these paths, it first considers those that appear to lead the fastest to the solution. The A* algorithm uses an evaluation function (Equation 7):(7)f(n)=g(n)+h(n)

The function f(n) represents the cumulative cost from the starting point to the current point, extending to the target point. Meanwhile, g(n) denotes the shortest cost from the initial point to the current position *n*, and h(n) predicts the optimal path cost from the current point n to the destination, often calculated as the Manhattan distance [64]. Initially applied in port areas, Casalino used the A* algorithm [65] for local pathfinding. Guan proposed an improved version of the A* algorithm [66], which helps Unmanned Surface Vessels (USVs) avoid static obstacles at sea and reach their destination smoothly while avoiding local minima. In addition, a collision-free trajectory planning method for space robots based on the A* algorithm has been developed in [67]. The geometric A* presented in [68] is designed for route planning of automated guided vehicles (AGVs) operating in port environments.

Despite its advantages, traditional A* does not always provide an optimal solution, as it does not take into account all feasible routes. In each iteration, A* evaluates the nodes based on their *f* values, which is a computationally expensive process, especially in large map search areas. Consequently, this approach can significantly slow down the speed of route planning.

### 3.2. Fuzzy Logic (FL)

Fuzzy logic (FL) is a technique for persuading the human intellect. FL is a uniform approximate (linguistic) method for inferring uncertain facts using uncertain rules [69]. In 1965, Lotfi A. Zadeh was the first to introduce the idea of an FL system [70]. The fuzzy sets he created are an extension of the traditional notion of a set, going beyond the Aristotelian (true–not–true; yes–no) division. The fuzzy set A is defined as follows [70]:

$$A=\{x,\mu_{A}\left(x\right)|x\in X,\mu_{A}\left(x\right):X\to \left[0,1\right]\}$$
where X is the so-called reference surface, and $mu_{A}\left(x\right)$ is the so-called membership function, which takes values in the complete closed interval between 0 and 1. In the special case where $mu_{A}\left(x\right)$ takes only values 0 and 1, *A* reduces to a classical set. The three basic operations on fuzzy sets (intersection, union, complement) are defined as extensions of the corresponding operations on classical sets. The standard properties of sets (De Morgan, absorption, associativity, distributivity, idempotence) hold here as well. Fuzzy inference (or fuzzy reasoning) is an extension of classical inference [69]. However, Zadeh’s vision was later expanded in several areas. The FL serves as a formal blueprint for representing and implementing the heuristic intelligence and observation-based methods of experts [71,72].

Figure 14 is an example of the primary FL driver used in [73]. The general architecture of a fuzzy logic controller consists of four units: IF–THEN rules, whose associated linguistic variable values can be not only true or false but can vary between the two; a fuzzy inference mechanism, which is a process to identify the output values associated with the input variables based on the fuzzy rules; an input fuzzification unit; and an output defuzzification unit. Hex Moor [74] was the first to apply the FL concept to robot path planning and obstacle avoidance. Since then, for example, the FL route planning approach has been applied in unknown environments [73]. Mobile robot routing algorithm based on FL and neural networks designed [75]. Chelsea and Kelly demonstrate FL controller for UAVs in a two-dimensional environment [6]. Then, 3D space navigation was demonstrated using FL for aerial [76] and underwater [77] robots and a Mamdani-type FL-based controller for a nonholonomic wheeled mobile robot that tracks moving obstacles [78].

### 3.3. Genetic Algorithm (GA)

A genetic algorithm is an optimization technique referring to genetics and natural selection, first introduced by Bremermann in 1958 [79]. It is based on Darwinian evolutionary theory and mimics the concept of survival of individuals best adapted to their environment. The most viable members of the population survive, while the weakest die off. The surviving members, depending on their fitness, allow the genes to be passed on to the next generation through cross-breeding, mutation, and selection. In this way, the individual fitness of the population continuously approaches the optimum. This random structure information was used to create a search algorithm that provided solutions to the problem of finding feasible pathways [79].

GAs stand for a sequence of algorithms. They randomly initialize populations with a character string and an objective function. Then, based on Darwinian evolutionary theory, they generate a new population using the three genetic operators (mutation, crossover, and selection). The new populations are created until the stopping conditions are met [50]. Such stopping conditions are a time limit, the required fitness value, and the maximum number of generations. During mutation, elements of an arbitrary string mutate with a given mutation probability. In a crossover, the elements of two strings are crossed according to a certain rule, thus creating two new strings. In selection, two strings selected by probability based on their objective function are compared based on their fitness, and the higher ranked higher-ranked one is selected to create the new population. The GA process is illustrated in Figure 15. The initial input comes from the population variables. This is followed by the encoding and decoding of chromosomes, the initialization of the population, and, the evaluation of the fitness values of the individuals within it. If the conditions are met, the optimal solution is obtained directly. Otherwise, the algorithm iterates, evolves, and selects new individuals from the population, whose fitness is re-evaluated until the condition is met. After that, the process stops.

GAs are used in many areas for mobile robot path planning problems, for example, for humanoid robot navigation [81], for the underwater robot navigation challenge in 3D route planning [82], and for aerial robots [83,84], as well as genetic-algorithm-based trajectory optimization for digital twin robots [85]. Work using improved genetic algorithms can be found in [86,87].

### 3.4. Simulated Annealing (SA) and Tabu Search (TS)

Simulated annealing and the Tabu search are approximate (heuristic) algorithms and therefore do not guarantee the optimal solution. They do not know when the optimal solution has been reached. Therefore, they need to be told when to stop. Easily designed to implement any combinatorial optimization problem, under some conditions, they converge asymptotically to the optimal solution. The same can be said for GAs [88].

#### 3.4.1. Simulated Annealing (SA)

SA is an iterative search method based on the analogy of annealing metals. Annealing is a process in which a low-energy state of the metal is created by melting the metal and then slowly cooling it. Temperature is the control variable in the annealing process and determines how random the energy state is [88]. Consider an energy diagram with two potential barriers. A ball is randomly placed on the potential curve and can only move down the curve. The ball then has an equal chance of going to *A* than to *B* (Figure 16).

Upward movements can be accepted at times with a probability controlled by the parameter temperature (T). For example, if you want the ball to move from pit *A* to pit *B* with a higher probability, you have to increase its temperature. The probability of accepting upward movement decreases as T decreases. At high temperatures, the search becomes almost random, while at low temperatures it becomes almost greedy. At zero temperature, the search becomes completely greedy, i.e., it accepts only downward movements. The algorithm is based on the Metropolis procedure, which simulates the heat treatment process at a given temperature T [89]. At the beginning of the procedure, the current temperature and solution are given, as well as the time for which the heat treatment at the given temperature should be maintained. The SA algorithm should start from a high temperature. However, if the initial temperature is too high, it will only result in a loss of time. The initial temperature should be such that virtually any proposed movement is acceptable, whether upward or downward. Thereafter, the temperature will gradually decrease. The annealing time increases as the temperature decreases. The annealing process stops when the time exceeds the permissible time [90].

The main part of the algorithm consists of two circles. In the inner circle, a possible move is generated and the acceptance of the move is decided by an acceptance function. The acceptance function assigns a $P_{accept}$ probability based on the current temperature and the cost change ΔC (Equation 8). At high temperatures, most uphill movements are likely to be accepted by the algorithm, regardless of the increase in costs. However, as temperatures fall, only downward movements are accepted. If the step is accepted, it is applied to the current path to generate the next state. The outer loop checks if the stopping condition is satisfied. Each time the inner loop completes, the temperature is updated using a function, and the stopping condition is checked again. This continues until the stop condition is met [90].
(8)$$P_{accept}=\left\{\begin{array}{c} e^{-\frac{\Delta C}{T}}\;\mathrm{if}\;\Delta C\geq 0\\ 1\;\mathrm{if}\;\Delta C<0 \end{array}\right.$$

#### 3.4.2. Tabu Search (TS)

TS is a combinatorial optimization technique that optimizes an initial given permutation or converts it to the closest possible optimal solution, by alternating successive steps. Using this method, it is possible to reduce the cost of a path by a series of edge swaps in a randomly generated round trip. The process continues until the path with the minimum cost is found. The selection of the best step to improve or not improve the current solution is based on the fact that good steps are more likely to reach the optimal or close to the optimal solution. The set of acceptable solutions in a given iteration forms a candidate list. The Tabu search selects the best solution from this candidate list, whose size reflects the trade-off between quality and performance. To Tabu the relocation attributes, a Tabu constraint is introduced to prevent the reversal of moves. This constraint is enforced by a Tabu list that stores the relocation attributes. The aspiration-level component allows the Tabu state to be temporarily overridden if the reversal results in a better solution than the best one achieved so far [88].

In [91], the design of minimal-cost delivery routes for goods-carrying mobile robots is developed using hybrid simulated annealing/Tabu search and approximation methods based on Tabu search algorithms, which start and end from a central warehouse while the robots serve customers. Each customer is supplied exactly once per vehicle path.

### 3.5. Particle Swarm Optimization (PSO)

Particle Swarm Optimization (PSO) is a nature-inspired approach that mimics the collective behavior of bird flocks, fish schools, or animal herds as they seek food, adapt to their surroundings, and interact with predators [92]. PSO draws inspiration from the foraging strategy observed in bird flocks, where individuals move towards the most favorable food sources guided by their knowledge, collective wisdom, and momentum. This behavior is emulated by the PSO algorithm through the representation of each potential solution as a particle, with personal and global best positions and inertia. Each particle maintains specific attributes such as position, velocity, and objective, striving to converge toward the global optimum over multiple iterations. The PSO process begins with the initialization of a randomly generated particle swarm, with each particle assigned a unique velocity to navigate the search space. Notably, unlike genetic algorithms, PSO assigns random weights to all potential solutions, enabling particles to explore the solution space dynamically. The algorithm’s functioning revolves around the interplay between particle positions and velocities, with each particle’s position updated based on its velocity conditions. Refer to Figure 17 for an illustration of this process [93,94].

Suppose that the search space is *D*-dimensional, and the *i*th particle of the population can be represented by a *D*-dimensional vector ${\left(x_{i}^{1},x_{i}^{2},\ldots ,xi^{D}\right)}^{T}$. The velocity of this particle can be represented by another *D*-dimensional vector ${\left(V_{i}^{1},V_{i}^{2},\ldots ,V_{i}^{D}\right)}^{T}$. The previously best-visited position of the *i*th particle is denoted by $P_{i}$, and the best particle in the swarm is denoted by $P_{g}$. The update of the particle’s position is accomplished by the following two equations: Equation (Equation 9) calculates a new velocity for each particle based on its previous velocity, and (Equation 10) updates each particle’s position in the search space [92,95].
(9)$$\begin{array}{c} V_{id}^{k+1}=wV_{id}^{k}+c_{1}r_{1}p_{id}^{k}t-x_{id}t+c_{2}r_{2}p_{g}^{k}t-x_{id}^{k}t\\ V\left(t+1\right)=wV\left(t\right)+\left[c_{1}r_{1}\left(P_{best}-x\left(t\right)\right)\right]+\left[c_{2}r_{2}\left(G_{best}-x\left(t\right)\right)\right] \end{array}$$
(10)$$\begin{array}{c} x_{id}^{k+1}t+1=x_{id}^{k}t+v_{id}^{k+1}+1\\ x(t+1)=x(t)+v(t+1) \end{array}$$
where *k* is the iteration number, d=1,2,3,…,D; i=1,2,3,…,N; and *N* is the swarm size. *w* is inertia weight, which controls the momentum of the particle by weighing the contribution of the previous velocity. $c_{1}$ and $c_{2}$ are positive constants, called acceleration coefficients. Alternatively, $c_{1}$ is also called the cognitive (local or personal) weight, and $c_{2}$ is the social (or global) weight. $r_{1}$ and $r_{2}$ are random values ranging from [0,1]. V(t) is the velocity associated with the particle at time *t*, and X(t) is the position of the particle at time *t*.

The PSO process, depicted in Figure 18, is characterized by rapid convergence but shows slower responses during particle search within a region. This limitation, due to its fixed convergence rate, can lead to localization issues [96].

PSO is widely applied in mobile robot path planning across various types, including humanoid [97], industrial, [98], wheeled [99], aerial [100], and underwater robots [101], particularly in complex three-dimensional environments.

### 3.6. Cuckoo Search Algorithm (CSA)

The concept of a cuckoo search is inspired by the behavior of cuckoo birds, which lay their eggs in the nests of other host birds (of different species). The cuckoo bird attempts to deposit its eggs in the nest of a host bird by removing one of the host’s eggs and replacing it with one of its own, which closely resembles the host bird’s eggs. Afterward, the cuckoo bird swiftly departs. The primary goal of this behavior is to safeguard its eggs from predators, as well as to ensure that its offspring have access to food and protection in the host nest. However, there is a risk that the host bird may detect the cuckoo egg and either remove it from the nest or abandon the nest to construct a new one. Consequently, the cuckoo continuously evolves its egg appearance to mimic that of the host bird’s eggs, reducing the likelihood of detection. Importantly, the host bird also learns to detect foreign eggs over time, perpetuating the cycle of egg-laying and detection. Once the cuckoo successfully places its egg in the host nest, a new phase ensues. Cuckoo chicks hatch earlier than the host bird’s offspring and may attempt to eject the host eggs or chicks from the nest. Additionally, cuckoo chicks compel the host mother bird to provide them with more food, potentially depriving the host chicks of sustenance altogether [102].

The interaction between the cuckoo and the host bird results in a direct conflict, as the host bird has a probability, denoted as P and ranging from 0 to 1, of detecting the cuckoo’s egg. If a host bird detects cuckoo eggs in its nest, it may either discard the egg or desert the nest altogether. These fundamental occurrences form the basis of the cuckoo search algorithm. The primary features of the CSA are outlined in [103]:

In the cuckoo search algorithm, a single egg is deposited by a cuckoo in a nest selected at random, symbolizing a potential solution to an optimization problem. The nest containing the most promising eggs—representing the optimal solutions—is carried forward to subsequent iterations. The total number of available nests remains constant, and each egg laid by a cuckoo is subject to a probability $\left(P_{a}\right)$ within the interval [0,1] of being detected and consequently abandoned. Consequently, during each iteration (t), a proportion $\left(P_{a}\right)$ of the entire population undergoes alteration.

The efficiency of the cuckoo search algorithm is enhanced through the utilization of Levy flight instead of random walk. Numerous animals and insects exhibit the characteristic Levy flight behavior. Levy flight entails a random walk with step lengths determined by a heavy-tailed probability distribution, as depicted in (Equation 11) [104]. Levy flight outperforms random walk in this regard. Hence, we opted for the cuckoo search algorithm in this research due to its ability to achieve faster convergence rates.
(11)$$\begin{array}{c} X_{i}\left(t+1\right)=X_{i}\left(t\right)+\alpha ⊕L\left(\lambda \right) \end{array}$$
(12)$$\begin{array}{c} \alpha =\alpha_{0}⊗\left(x_{i}\left(t\right)-x_{best}\right) \end{array}$$
where $X_{i}\left(t+1\right)$ represents the new solution, *t* indicates the current generation (iteration) of the solution, α is the step-wise parameter that controls the moving step size of the cuckoo, ⊕ is entry-wise multiplication, and $\alpha_{0}$ denotes the step size factor, which is usually set to 0.01. and L(λ) is Levy exponent, which stands for a random search path, which can be expressed as
(13)$$L\left(\lambda \right)=\frac{\varphi \times m}{{\left|n\right|}^{\frac{1}{\beta }}}$$
where *m* and *n* are two random numbers subjected to the normal distribution, β is set to 1.5. φ is defined as:(14)$$\varphi ={\left(\frac{\Gamma \left(1+\beta \right)\times sin\left(\pi \frac{\beta }{2}\right)}{\Gamma \frac{1+\beta }{2}\times \beta \times 2^{\frac{\beta -1}{2}}}\right)}^{\frac{1}{\beta }}$$

Algorithms with high computational complexity typically demand significant resources, which may not always be feasible. The cuckoo search algorithm (CSA), however, requires only a few initial parameters, enabling efficient resolution of multimodal problems. The CSA, depicted in Figure 19, involves three key operations: (i) Levy flight for generating new solutions, (ii) replacement of nests with superior solutions based on fitness evaluations, and (iii) greedy selection to maintain the best solutions until the goal is achieved.

CSA has been effectively hybridized with an adaptive neuro-fuzzy inference system for enhancing the navigation of multiple mobile robots in unknown environments [105] and applied in vehicle track design [106] and scheduling [107]. Additionally, it has been used in a novel artificial neural network approach to predict ground vibrations from mine blasting [108].

### 3.7. Artificial Bee Colony (ABC)

Karaboga developed the Artificial Bee Colony (ABC) technique, a swarm-based algorithm inspired by the foraging behaviors of bees [109]. The three rules of the ABC model are as follows: (a) Forager bees: Forager bees are sent to the food sites (the nearest colony) and inspect the quality of the food. (b) Inactive forager bees: Based on information from active forager bees, inactive bees inspect the food sources detected and assess/assess them. (c) Food sources: Forager bees that find rich food sources distribute them, while forager bees with few food sources give them up, creating a problematic situation.

The population is initialized from the set of employed and onlooker bees. Each worker is sent to the food source ($x_{i}^{j}$) that the bee is responsible for, and according to (Equation 15), the fitness value of each source of food is determined [110].
(15)$$fit_{i}=\left\{\begin{array}{c} \frac{1}{1+f_{i}}\;\mathrm{if}\;f_{i}\geq 0\\ 1+\left|f_{i}\right|\;\mathrm{if}\;f_{i}<0 \end{array}\right.$$
where the objective function $f_{i}$ shows the fitness value of source $x_{i}$. In addition, the failure counter, which is a limit value for each food source, is defined and initialized to zero.

Then, using (Equation 16), they try to find a better food source ($v_{i,j}$).
(16)$$v_{i}^{j}=x_{i}^{j}+\alpha_{i}^{j}\left(x_{i}^{j}-x_{k}^{j}\right)$$
where j=1,2,⋯,D. The problem dimension is defined by *D*. k=1,2,⋯,N. N represents the total number of employed or onlooker bees. The value of *k* is not equal to *i*, and $\alpha_{i}^{j}$ is a random number generated from a uniform distribution in [−1,1].

Should the fitness value of the new position surpass that of the current one, the bee retains the newly identified food source location and disregards the previous source. The worker bee then communicates the fitness value of this new food source to the onlooker bee. The onlooker bee evaluates each food source based on the probability $P_{i}^{j}$ and selects the optimal food source $x_{i}$. The probability evaluation of the food source is determined using Equation (Equation 17) [110].
(17)$$P_{i}^{j}=\frac{fit_{i}}{\sum_{j=1}^{N}fit_{j}}$$
where $fit_{i}$ is the fitness value of the solution *i*. This is clear from the ABC algorithm’s general flow chart (depicted in Figure 20) [111].

The ABC algorithm has been used to solve many real-world problems. Ref. [112]’s applications of the ABC algorithm can be seen in many situations where MR (moving robot) systems operate in static environments [113,114]. For example, they tested a wheeled MR underwater [115], applying it to the routing problem of autonomous vehicles [116], as well as aerial robots [117]. The modified ABC algorithm was used for the Unmanned Combat Aerial Vehicle (UCAV) navigation problem [118] to plan optimal routes in a three-dimensional environment, including unmanned helicopters [119].

### 3.8. Ant Colony Optimization (ACO)

This algorithm is inspired by the foraging behavior and communication of ants, and it was presented by Dorigo and Maniezzo in 1991 [120]. Ants leave behind a kind of pheromone on the paths they traverse. The more ants travel along a path, the more pheromone accumulates on it, and other ants will follow stronger pheromone trails left by other ants in the area. When an ant initiates a search process in a problem, for example, searching for a route on a map, it randomly selects a route and follows it. As it progresses, the ant senses the amount of pheromones in the environment and makes decisions to modify its route based on this information. Ants prefer routes with higher pheromone concentrations. The ACO algorithm runs repeated colonies of ants and compares the results of each colony to optimize the pathways based on the amount of pheromones. In this process, the algorithm gradually converges to an optimal solution to the problem. The formulae of the ACO algorithm (Equation 19) are described in [80]:(18)$$\begin{array}{c} P_{ij}^{k}\left(t\right)=\left\{\begin{array}{c} \frac{\tau_{ij}^{\alpha }\left(t\right)n_{ij}^{\beta }\left(t\right)}{\sum_{s\in d_{k}}\tau_{ij}^{\alpha }\left(t\right)n_{ij}^{\beta }\left(t\right)}\;s\in d_{k}\\ 0\;\mathrm{otherwise} \end{array}\right.\\ n_{ij}=\frac{1}{d_{ij}} \end{array}$$
where $P_{ij}^{k}\left(t\right)$ is the transition probability, $\tau_{ij}^{\alpha }\left(t\right)$ represents the pheromone concentration, $n_{ij}^{\beta }\left(t\right)$ is the heuristic function, $d_{k}$ is a collection of access points, and $n_{ij}$ is a heuristic function, usually expressed as the reciprocal of the distance $d_{ij}$ between *i* and *j*.
(19)$$\begin{array}{c} \tau_{ij}\left(t+\Delta t\right)=\left(1-\rho \right)\tau_{ij}\left(t\right)+\Delta \tau_{ij}\left(t\right) \end{array}$$
(20)$$\begin{array}{c} \tau_{ij}=\tau_{ij}+\sum_{k=1}^{M}\Delta \tau_{ij}^{k} \end{array}$$
(21)$$\begin{array}{c} \Delta \tau_{ij}^{k}=\left\{\begin{array}{c} \frac{1}{d_{ij}}\;\mathrm{Ant}\;k\;\mathrm{pass}\;\left(i,j\right)\\ 0\;\mathrm{otherwise} \end{array}\right. \end{array}$$
where ρ represents the pheromone volatility coefficient, *M* is the total number of ants in the ant colony, and $\Delta \tau_{ij}^{k}$ represents the pheromone amount released by the *k*th ant. The ACO process is illustrated in Figure 21.

Initially applied to solving the Traveling Salesman Problem (TSP) [120], the principles and mathematical models of the ACO algorithm have since been systematically studied and have undergone significant development, such as in [121] with airport AGV route optimization model based on the ant colony algorithm for optimizing Dijkstra’s algorithm in urban systems. In [122], a search and rescue is presented in a maze-like environment with ant and Dijkstra algorithms. The work in [123] describes the application of odometry and Dijkstra’s algorithm to warehouse mobile robot navigation and shortest path determination.

### 3.9. Deep-Learning-Based Control (DL)

Machine learning (ML) is the process of using computer systems to learn and improve without their experience, explicitly programming them. Machine learning algorithms rely on recognizing patterns and rules from data and making decisions or predictions based on them. Basic machine learning techniques include supervised learning (where algorithms are trained on labeled data), unsupervised learning (where algorithms try to find structure from unlabeled data), and semisupervised learning, which uses a combination of the two methods. Deep learning is a specialized field of machine learning that uses deep neural networks to learn complex patterns and representations. Deep learning enables computer systems to learn representations of data using multilayered, hierarchical structures. These layers gradually learn higher-level features, which makes deep learning algorithms particularly effective in image recognition, speech recognition, natural language processing, and many other complex tasks. Deep learning models such as Convolutional Neural Networks (CNNs) and Recurrent Neural Networks (RNNs) have made significant breakthroughs in various application areas of artificial intelligence. The main advantage of solutions based on machine learning is that they can learn from the data, so their models already incorporate the nonlinear behavior of the control plant. This enables better performance in many control applications than classical approaches. Deep learning techniques are suitable for handling both global and local path-planning problems [124].

#### 3.9.1. Artificial Neutral Network (ANN)

A neural network, which draws inspiration from the natural human senses, serves as an intelligent system and was originally devised for mobile robot route planning [125]. It consists of simulated networks composed of neuron-like units. These networks undergo optimization through comprehensive training on designated tasks, with the connection strengths between units being gradually adjusted over time [126]. In a neural network, the processing elements (neurons) are usually ordered topologically and interconnected in a well-defined way. The structure of the neural network plays an important role in the execution of the task. Due to the internal parallel structure of neural nets, computations can be performed in parallel, thus ensuring high processing speed. Thus, neural networks are particularly suitable for solving real-time tasks.

A general neuron structure is shown in Figure 22.

Where $x_{i}$ is input to the neuron, $X=[x_{1},x_{2},\ldots x_{n}]$ is the input vector (*n* represents the number of inputs on the neuron). *b* is a constant input (bias-offset value), and *y* is the neuron’s output. $w_{i}$ represents the weight factor associated with the *i*th input, $W=[w_{1},w_{2},\ldots w_{n}]$ is the weight vector, and *f* represents the activation function.

The $x_{i}$ scalar inputs are summed by weighting $w_{i}$ and the weighted sum is then summed to a nonlinear element. The weighted sum of the input signals, which is the input to the activation function, is called the excitation, while the output signal is called the response (activation). The *f* function is called the activation function. The output of a neuron can be calculated as follows:(22)$$y=f\left(\sum_{i=1}^{n}x_{i}w_{i}-b\right)$$

The weight factors determine the degree of influence on connections with neighboring neurons within a neuron’s vicinity. A neural network’s functionality relies on these weight factors, which encapsulate information or the processing of information during the learning phase.

Utilizing a nonlinear activation function enables the neural network to model any nonlinear function when applied to a suitable neuron. Conversely, a linear activation function leads to a linear neural network. To imbue a neural network with nonlinearity, it is imperative to incorporate at least one nonlinear activation function. Additionally, differentiation plays a crucial role, as gradient-based learning stands as the predominant method for adjusting neural network weights.

ANNs are structured into distinct layers: the input layer, where known data are fed into the model; the intermediate layers, referred to as hidden layers; and the output layer, which yields the final sought-after value. Each layer comprises various units (neurons or nodes), with each unit connected to the subsequent layer through a transfer function. Within an ANN, the output of layer i−1 serves as the input for layer *i*. The known data enter the input layer, accompanied by a bias term. Subsequently, these data are subjected to multiplication by initial weights, followed by summation. The resulting values are then passed through functions to the subsequent layer, iterating until reaching the output layer, where the final value is derived. Transitioning from one layer to the next in an ANN involves the utilization of transfer functions [127]. A possible layout of an artificial neural network is illustrated in Figure 23.

The operation of neural networks can typically be divided into two phases:Learning phase—the network stores the desired information processing procedure in some way.Recall phase—the stored procedure is used to execute information processing.

The main forms of learning in neural networks [128]:Learning with a teacher (called supervised or guided learning (also known as controlled learning)).Reinforcement learning.Learning without a teacher (unsupervised or unsupervised learning).Analytical learning.

Learning neural networks is nothing more than a multivariate optimization procedure based on a predefined criterion function (cost function). Various optimization techniques have been widely used for learning neural networks: gradient-based strategies [129,130], evolutionary methods, genetic algorithms [131,132], and particle swarm optimization (PSO; see later) algorithms [133].

ANN has been applied in a wide range of fields, including search optimization [134], pattern recognition [135,136], image processing [137,138], mobile robot routing [139], signal processing [140], and many more. A hybrid approach to mobile robot navigation combining an ANN and FL [141,142] was designed for a mobile robot navigation controller using a neuro-fuzzy logic system.

In [143], a single-layer approach to robot tracking control was proposed. Through experiments on a KUKA LBR4+ robotic manipulator, the feasibility of the novel ANN approximation for robot control was examined. Ref. [144] presents positioning the error compensation of an industrial robot using neural networks. Ref. [145] presents a recurrent neural network for prediction of motion paths in human robot collaborative assembly.

The ANN was extended to create the Guided Adaptive Pulse Coupled Neural Network (GAPCNN) for mobile robots [146]. The GAPCNN aims to achieve fast parameter convergence to help the robot move in both static and dynamic environments. In particular, the ANN method has been used in MATLAB for mobile robot trajectory planning problems for aerial robots [147], humanoid robots [148], underwater robots [149], and industrial robots [150].

#### 3.9.2. Model Predictive Control (MPC)

The MPC method (Figure 24) is used to predict the behavior of the system for a given time interval and, based on the prediction, optimize the intervention signal at each time instant. As a result, it minimizes the cost function and determines the optimal control sequence. The method has the advantage of a user-friendly design process and easy implementation. It has many applications in the automotive industry, for example, it is used to solve tracking problems [151]. The path planning of autonomous vehicles can also be conducted using a predictive approach [152,153].

Due to the high computational complexity of numerical optimization, it is of paramount importance to ensure real-time computability, which requires the right formulation of the problem and the choice of the appropriate procedure for its solution. The most commonly used MPC approach is based on linear models, but this also has limitations that can reduce performance. Advances in recent decades have allowed engineers to use control approaches that have a higher computational cost, such as Nonlinear Model Predictive Control (NMPC). The main drawback of an NMPC is its complexity, which can lead to high computational time. As a consequence, in most cases, only suboptimal solutions can be obtained, which may degrade the performance of the closed-loop system [154].

Ref. [155] also proposes a cooperative regulatory strategy for docking unmanned aerial vehicles (UAVs) based on MPC. The proposed strategy implements a nonlinear and a linear MPC for the coarse approach (long range) and the fine docking maneuver (short range) based on the same objective function with tailored optimization strategies. Docking is a complex, critical maneuver that requires knowledge of the flight safety of the docking route and the constraints associated with the position of the platform to be docked. In addition, nonlinear effects such as vorticity due to the close approach of the lead agent must be taken into account.

Using the MPC method, it is easier to prove the stability and performance of the system, as it does not require knowledge of the system model. Instead, a local model is used and updated every time step. In addition, other methods are available to redefine the learning characteristics compared with neural networks. For example, in [156], an MPC-based control solution is proposed where the terminal cost and the set are determined through an iterative process.

The presented algorithms do not guarantee stability, which makes their application in safety-critical systems, such as autonomous vehicles, risky. However, some solutions address this problem. For example, ref. [157] presents a control strategy based on safety settings that can modify the input signal of the system when the output of a machine learning agent may destabilize the system. Another solution is given in [158], in which a Hamilton–Jacobi reachability algorithm is exploited that can work with any machine learning-based solution. A combined approach is presented in [159], in which a classical controller is used to control the linearized system, while the machine learning-based algorithm handles the nonlinearities of the system.

B. Németh [160] incorporated machine learning into the usual model-based robust control theory framework, but emphasized it as a new tool and an additional data-driven branch. For all its learning nature, however, robust control remains, i.e., the traditional model-based solution has been extended to fit today’s new approach to new types of tasks. The method is independent of the internal structure of the learning-based control element. Hence, a control element with any structure can be incorporated in its place, providing considerable freedom in control design. For example, solutions based on neural networks, which are already well established in practice, can be implemented in the developed control solutions for reference signal training or feedback loops. However, data-based MPC-type control schemes typically have a more closed, less flexible formalism for the optimization formulated in them. Another consequence of the hierarchical structure is that the learning-based management element can be physically separated from the supervisor and robust management elements. The vehicle motion dynamics are considered in the robust control element and the learning functions in the learning-based control element. For example, in the context of automated vehicles, the supervisor-robust control dual, which has a low computational demand, can be placed on board the vehicle, while the learning-based control element can be placed on an independent platform, such as a cloud. Control solutions that rely predominantly on solving an optimization task online typically do not have this advantage. The supervisor element, which requires online computation, has significantly lower computational requirements than traditional MPC or more advanced data-based (learning) MPC solutions. This is because the supervisor performs significantly fewer tasks than the main optimization task of the MPC. In the supervisor, it is not necessary to perform an optimization over a long horizon, since the impact on future motion states is taken into account in the learning process by running on episodes or prespecified patterns.

#### 3.9.3. Deep Reinforcement Learning (DRL)

Reinforcement learning (RL), inspired by animal psychological learning, learns optimal decision-making strategies from experience [161]. RF is a special type of ML algorithm that does not require large amounts of data for training. The RF algorithm is modeled based on reward, and several papers address the problem of autonomous vehicle control using RF methods [162,163]. Although RF-based solutions can be efficient, the stability of the closed-loop system is still an open question. A proposed solution is an RF-based algorithm combined with a robust controller [164], which achieves the stability of the algorithm by applying uncertainty models. The Deep Reinforcement Learning (DRL) model is particularly promising for solving Vehicle Routing Problems (VRPs). DRL can estimate patterns that are difficult to find with manual heuristics, especially for large-scale problems. Moreover, DRL can generate and infer routes quickly, making it extremely useful for solving time-sensitive VRPs.

The use of DRL in mobile robot navigation is a growing trend. The purpose of using the DRL algorithm in an autonomous navigation task is to find the optimal policy for guiding the robot to the target position through interaction with the environment. The advantage of DRL-based navigation is that it is map-free, has strong learning ability, and has little dependence on sensor accuracy [124].

### 3.10. Other Algorithms

Without wishing to be exhaustive, we briefly mention some algorithms that have recently become common.

#### 3.10.1. Dynamic Window Approach (DWA)

The DWA can generally be classified as a heuristic method, as it does not rely on rigorous mathematical models or algorithms to solve the problem but rather on an empirical approach. This method is designed for local routing and obstacle avoidance [165]. It takes into account the robot’s current speed, acceleration limits, and immediate surroundings to calculate a safe and feasible path to the destination. Creates a dynamic window based on possible velocities and angular velocities. An objective function calculates the optimal value of these pairs of velocities based on the minimum distance from the obstacles, the final bearing angle, and the velocity values of the robots. While in less complex environments the DWA can deftly avoid obstacles, its performance in extremely crowded environments may be suboptimal [166]. The DWA has the local minima and the global convergence problems [167].

Once the task creator has set one or more targets and the global route has been planned, the execution phase involves the robot scanning the surrounding environment, planning local trajectories, and moving forward. This sequence is repeated until the goal is reached. At the beginning of this flow, it samples all the speed pairs corresponding to the kinematic constraints of the robot. DWA computes the coordinates of the waypoints for each input velocity pair using iterations (Equation 23) to (Equation 25) [168].
(23)$$x\left(t_{n}\right)=x\left(t_{n-1}\right)+v\cdot \Delta t\cdot \cos\left(\Theta \left(t_{n-1}\right)\right)$$
(24)$$y\left(t_{n}\right)=y\left(t_{n-1}\right)+v\cdot \Delta t\cdot \sin\left(\Theta \left(t_{n-1}\right)\right)$$
(25)$$\Theta \left(t_{n}\right)=\Theta \left(t_{n-1}\right)+\omega \cdot \Delta t$$

The model assumes that the robot moves a distance of v·Δt along the heading of $t_{n-1}$ and then rotates an angle of ω·Δt, where x(t) and y(t) represent the coordinates, and Θ(t) represents the heading of the robot. By iterating the input velocities, this method computes the coordinates and heading of the robot from time $t_{0}$ to $t_{n}$. The computation time depends on the number of iterations. In the next step, DWA calculates the distance between each obstacle and waypoint using matrix operations. The calculation time is influenced by the number of paths and obstacle points. Subsequently, DWA swiftly determines the direction to the path’s endpoint and the distance to the target. After assessing all possible speed pairs, the optimal speed command is generated. This model is extensively utilized in research involving wheeled robots [168].

#### 3.10.2. Golden Jackal Optimization (GJO)

GJO is a metaheuristic, swarm-intelligence-based algorithm proposed by Nitish Chopra and Muhammad Mohsin Ansari, which models the cooperative hunting behavior and tactics of golden jackals in nature. Because these opportunistic animals are famous for their ability to adapt to different environments [169]. Golden jackals usually hunt with males and females. After finding the prey, they begin to move towards it cautiously. The prey is then surrounded and stalked until it stops. Finally, it is attacked and captured. Updating the position of the prey often depends on the male golden jackal. For this reason, the diversity of golden jackals is not adequate in some cases, and the search algorithm tends to fall into the local optimum [170].

GJO initiates with a randomized distribution of the first solution across the search space, as shown in Equation (Equation 26) [169]:(26)$$Y_{0}=Y_{min}+rand\left(Y_{max}-Y_{min}\right)$$
where $Y_{max}$ and $Y_{min}$ are the maximum and minimum values of the variable *Y*, and $rand(Y_{max}-Y_{min})$ is a uniform random vector in the range of 0 to 1.

In [171], a hybrid-strategy-based GJO algorithm for robot path planning is presented.

#### 3.10.3. Grey Wolf Optimization (GWO)

GWO is another type of swarm intelligence algorithm [172] that mimics the hunting strategy of wolves. It categorizes the wolves into different roles: the chief wolf, α, who leads the hunt; β, who assists the leader; δ, who scouts and guards; and the rest ω. The wolves’ hunting process is generally broken down into three phases: encirclement, pursuit, and attack. During the encirclement phase, the algorithm updates positions using Equation (Equation 27):(27)$$X\left(t+1\right)=X_{P}\left(t\right)-A\left|C\cdot X_{P}\left(t\right)-X\left(t\right)\right|$$

Although GWO is efficient, it needs a unique initial population. Another drawback is its slow convergence and easily falling into a local optimum [173]. Shitu Singh [174] proposed a more advanced version using Levy’s flight model to modify the population and the greedy selection method to update the path.

The Grey Wolf Optimization algorithm has been successfully applied to route planning [175]. The Golden Sine Grey Wolf Optimizer (GSGWO) has been improved from the Grey Wolf Optimizer (GWO), which provides slow convergence speed and easily falls into local optimum, especially without an obstacle-crossing function [176].

#### 3.10.4. Gravitation Search Algorithm (GSA)

GSA is also a robust metaheuristic population-based search algorithm based on gravity rules [177]. Objects are attracted to each other by the force of gravity, and this force is responsible for the global movement of all objects towards more massive objects. The masses thus interact through gravitational force. Heavy masses, which are good solutions, move more slowly than lighter masses (bad solutions). The position of the mass corresponds to the solution of the problem. The gravitational and inertial mass of bodies is determined by a fitness function. GSA can be seen as an isolated system for masses.

Like other metaheuristic systems, GSA has parameters that greatly affect its performance. The mass *j* acting on mass *i* by mass *j* is the equation $F_{ij}$ giving the gravitational force and the gravitational acceleration $a_{i}$ caused by it (Equation 28) [177]:(28)$$\begin{array}{c} F_{i,j}=G\frac{M_{aj}M_{pi}}{R^{2}} \end{array}$$(29)$$\begin{array}{c} a_{i}=\frac{F_{i,j}}{M_{ii}} \end{array}$$
where $M_{aj}$ and $M_{pi}$ represent the active gravitational mass of particle *i* and passive gravitational mass of particle *j*, respectively, *R* is the distance between masses, and $M_{ii}$ represents the inertia mass of particle *i*. “G(t) is the gravitational constant that decreases iteratively” [178]:(30)$$G\left(t\right)=G_{0}e^{-\alpha \frac{t}{T}}$$

The gravitational constant *G* is the most sensitive entity in the GSA model and effectively controls the balance between the exploration and exploitation capabilities of the algorithm. α and $G_{0}$ are constant parameters that affect the performance of the algorithm. As for the tuning of the mentioned parameters, many GSA variants have been developed [178].

## 4. Hybrid Algorithms

As you can see, there are many other ways to avoid obstacles. Among these are many that use several classical or heuristic algorithms at the same time, which are also described in this article. These are commonly referred to as “hybrids”. In this article, three such algorithms are mentioned as an addition.

### 4.1. New Hybrid Navigation Algorithm (NHNA)

The so-called “new hybrid navigation” algorithm consists of two independent layers, the deliberative and reactive layers. The deliberative layer plans the reference route using the A* search algorithm based on the stored preliminary information. The reactive layer takes over the reference trajectory and guides the robot autonomously along the planned route [25]. The reference path is temporary, and it can be changed by the reactive layer during movement. This layer uses the D-H error algorithm (Distance Histogram bug). It is an improved version of the bug-2 algorithm [42], which allows the robot to freely rotate at angles less than 90° to avoid obstacles. If a rotation of 90° or more is required to avoid an obstacle, the bug-2 algorithm behaves as a bug [44]. The algorithm needs prior information about the environment, which it stores as a binary grid map. The state of each grid on the map is free or occupied: free if there is no obstacle in it, and occupied if it has an obstacle. Figure 25 shows the results of [25], which shows the planned and shortest paths generated by the algorithm. Figure 26a shows the path of the robot with the Dist-Bug algorithm, while Figure 26b shows the behavior of the robot with the D-H error algorithm [25].

### 4.2. Hybrid Navigation Algorithm with Roaming Trails (HNA)

This algorithm is designed to effectively handle environments where the robot encounters obstacles during movement. During navigation, the robot can deviate from its path to avoid obstacles using reactive navigation strategies, but is always limited within the area. By ensuring the robot moves within a convex area encompassing the target node’s location, it is assured to reach the target in the presence of static obstacles by following a straight path. In certain scenarios, the mobile robot must navigate around obstacles or come to a halt when faced with an obstacle. [44]. The main difference between the hybrid navigation algorithm and NHNA is that it uses APF instead of D-H BUG in the reactive layer. NHNA did not describe any constraints on the deviation from the reference path, but HNA used the concept of roaming trails for the same purpose. Figure 27 shows the roaming traces with the preliminary map (top) and the safe trajectory of the robot on the roaming traces (bottom) [179].

For more than ten years, the approach has been extensively tested on robots, in particular on the autonomous robot Staffetta [179]. Staffetta is specifically designed for autonomous transport in hospitals, with a payload of 120 kg and a maximum speed of 1 m/s. The robot is equipped with sensors to detect nearby objects and touch sensors to avoid collisions. Furthermore, it is equipped with a laser-based localization system that allows regular position corrections. Based on the experimental experience gained, the second generation of the robot (Merry Porter™) has been further developed and is now independently transporting waste within the Modena Polyclinic.

### 4.3. Methods Based on Sliding Mode (SM)

The intelligent space learns motion control by tracking the robot’s movements [180], thus being able to learn an obstacle avoidance strategy. This learning is based on a neuro-fuzzy approximation of vector-field-based obstacle avoidance. The efficiency of navigation is crucial, as the main application tasks of a mobile robot may include, for example, the guidance of visually impaired people, which requires an immediate reaction to any disturbance. Using the artificial potential field, a collision-free trajectory is guaranteed along gradient lines. The equations of motion of the robot concerning the fixed world system $\left(x_{f},y_{f}\right)$ can be derived as follows:(31)$$\begin{array}{c} \dot{x}=v_{G}\cos\phi \\ \dot{y}=v_{G}\sin\phi \\ \dot{\phi }=v_{G}/L\tan\theta \end{array}$$
where $v_{G}$ denotes the velocity vector at the center of the moving platform, which is constrained along the longitudinal axis fixed to the robot due to nonholonomic kinematics. In the robot fixed coordinate system $\left(x_{R},y_{R}\right)$, a local harmonic potential field Ψ(x,y) is generated [181]. According to Laplace’s equation, this harmonic field corresponds to
(32)$$\nabla^{T}\cdot \nabla \Psi \left(x,y\right)=\frac{\partial^{2}\Psi \left(x,y\right)}{\partial x^{2}}+\frac{\partial^{2}\Psi \left(x,y\right)}{\partial y^{2}}=0$$

The solution to (Equation 32) gives the potential of a singular point of power *q* at (0,0) in a 2D Cartesian (x,y):(33)$$\Psi \left(x,y\right)=q\ln\frac{1}{\sqrt{x^{2}+y^{2}}}$$
and the associated gradient $\rho \left(x,y\right)\in R^{2}$:(34)$$\rho \left(x,y\right)=-grad\Psi \left(x,y\right)=\frac{q}{\sqrt{x^{2}+y^{2}}}\left(\begin{array}{c} x\\ y \end{array}\right)$$

The configuration of the fundamental potential field consists of a negative unit singular point in the target and a positive singular point of magnitude 0<q<1 in the middle of the obstacle, $q=\frac{R}{R+e}$, where *e* is the distance between the target and the center of the obstacle, and *R* is the radius of the circular safety zone. As circular obstacle protection zones cannot be applied directly [181], elliptical safety zones have been designed. Each obstacle has one safety ellipse, but if there are multiple obstacles, two ellipses are needed on either side of the selected route. In this case, the two potential fields must somehow be “merged” to form a single potential field. A good alternative method is to always consider only the nearest safety ellipse. However, this requires switching potential fields at the intersection of equidistant lines between ellipses. This switching, as shown in Figure 28, results in a noncontinuous gradient field.

In this case, the sliding surface can be described by the line $\sigma_{eq}=0$. When switching between gradient lines, the scattering appears as oscillations. This effect can be reduced by smoothing the gradient lines near the equidistance line: in the boundary layer along the equidistance line between the two safety zones by spatial domain smoothing. The gradient of the resulting smooth gradient field is the weighted sum of the two gradients. The control inputs are usually the outputs of some actuator. The gradient ρ(x,y) is implemented as a velocity field. Kinematics constrains robot motion from three-dimensional to two-dimensional along the velocity vector. We assume that the state variables x,y,ϕ and the kinematic parameters L and W are known. The orientation of the robot’s angle ϕ must be controlled to be colinear concerning the gradient ρ(x,y). So the desired orientation at the point (*x*, *y*) is [181]:(35)$$\phi_{\rho }=Atan\frac{\rho_{y}}{\rho_{x}}\;with\;\rho \left(x,y\right)=\left(\begin{array}{c} \rho_{x}\\ \rho_{y} \end{array}\right)\in R^{2}$$

Because speed control is simple, the desired direction of movement β, v=β|v|, where β is defined by the orientation error ▵ϕ, as follows:(36)$$\begin{array}{c} ▵\phi =\phi_{\rho }-\phi +2\pi \Rightarrow \beta =1\;ha\;-2\pi <\phi_{\rho }-\phi <-\frac{3\pi }{2}\\ ▵\phi =\phi_{\rho }-\phi +\pi \Rightarrow \beta =-1\;ha\;-\frac{3\pi }{2}<\phi_{\rho }-\phi <-\frac{\pi }{2}\\ ▵\phi =\phi_{\rho }-\phi \Rightarrow \beta =1\;ha\;-\frac{\pi }{2}<\phi_{\rho }-\phi <\frac{\pi }{2}\\ ▵\phi =\phi_{\rho }-\phi -\pi \Rightarrow \beta =-1\;ha\;\frac{\pi }{2}<\phi_{\rho }-\phi <\frac{3\pi }{2}\\ ▵\phi =\phi_{\rho }-\phi -2\pi \Rightarrow \beta =1\;ha\;\frac{3\pi }{2}<\phi_{\rho }-\phi <-2\pi \end{array}$$

The sliding surface of the orientation error is defined as follows:(37)σ=β▵ϕ

A sliding mode along the σ=0 surface is created, but at the same time, the direction of motion is changed, and changing the sign of β should be avoided. This can be avoided by monotonically decreasing ▵ϕ by controlling the value of ϕ [181]. The Lyapunov function in this case is $V=\frac{1}{2}\sigma^{T}\sigma$. Differentiating this function along the trajectories of the system:(38)$$\sigma^{T}\dot{\sigma }=\sigma^{T}v\left(S\cos\theta -\frac{1}{L}\right)$$
where S(x,y,ϕ) describes the rate of change in the curvature of the gradient along the track lines, ϕ=arctanSL, and the θ is
(39)$$\theta =\varphi +\frac{\pi }{2}sign▵\phi$$

### 4.4. Other Examples

Just a few more examples of hybrid algorithms are provided below:Ref. [182] presents a hybrid path-planning algorithm based on improved A* and an artificial potential field for unmanned surface vehicle formations.Researchers have also exploited GA hybridization with other approaches to MR navigation for better results in route planning problems, such as GA-PSO [183], GA-FL [184], and GA-ANN [185].In [186], a hybrid genetic algorithm (HGA)-based approach applied to the image denoising problem is presented. HGA provides the dynamic mutation rate and a switchable global-local search method for the mutation operator of the ordinary genetic algorithm [187].In [188], the dynamic modeling of the impact of polymer insulators in polluted conditions based on the HGA-PSO algorithm is presentedRef. [189] used a Voronoi diagram and the particle swarm optimization algorithm to achieve multirobot navigation and obstacle avoidance.Ref. [190] presents a UGV routing algorithm based on an improved A* with an improved artificial potential field.Ref. [43] used VFH*, combining the VFH+ local obstacle avoidance algorithm and the A* path planning algorithm.

## 5. Comparison of the Algorithms Discussed in This Paper

The advantages and disadvantages of the classical and heuristic algorithms and the convergence and computation time requirements are summarized in Table 1, Table 2, Table 3, Table 4 and Table 5. Convergence time and computation time are expressed in different units and scales. Data are approximate values only and may vary depending on circumstances.

## 6. Discussions and Future Trends

Navigation and route planning are the central difficulties of mobile robots and have been the subject of decades of research. As a result, several methodologies have been presented and applied to the problem of route planning for mobile robots. Strategies for mobile robot optimization can be classified into deterministic or classical approaches and nondeterministic or heuristic approaches. Traditional algorithms execute a given task step by step according to predefined instructions, and their results are exact and deterministic. (One of the simplest algorithms is the Pythagorean theorem, which determines a third parameter in an identical way given two input parameters, and the term heuristic is derived from the Greek word heuresis, which means to find.) Theoretically, constructing an exact solution procedure would make it possible to calculate how to reach the goal based on the robot’s current position and by analyzing all possible paths. The problem is that the situation becomes too complex above a given complexity of the environment. A heuristic algorithm does not consider all possible steps but only decides according to some logic based on a particular part of the problem space. A considerable advantage of heuristic algorithms is that they can deliver results relatively quickly for high-complexity problems with little computation. However, they have the disadvantage that the optimal solution cannot be guaranteed completely. They are helpful when the solution to a problem cannot be found within a foreseeable time by a conventional method that provides an exact solution. They can also provide an optimal or approximate solution for large problem sizes. Among the earliest developed error avoidance algorithms, they are straightforward to calibrate but time-intensive. These methods are not goal-oriented; they trace edges without considering the ultimate objective [37]. The Dijkstra algorithm is a graph search algorithm designed to find paths and determine the shortest paths [5]. The Floyd-Warshall (FW) algorithm uses a weighted and directed graph and can compute opposing weighted edges. The solutions are derived from the previous results, and multiple solutions can be generated [191]. This algorithm finds the shortest path between each pair of nodes and is particularly useful when the distance between all pair nodes of the graph needs to be determined. However, it is inefficient for large graphs due to its high memory and time requirements. The Bellman-Ford (BF) algorithm can find the shortest path from one peak to another, which is simple and does not require complex data structures to apply. The algorithm iteratively extends the search to all nodes, not just along the current shortest path. For this reason, it can be slower than Dijkstra’s algorithm for positive-weight graphs but can handle graphs with negative weights, whereas Dijkstra’s algorithm cannot. If there is a negative cycle in the graph, the Dijkstra algorithm would run the cycle infinitely, as this would theoretically result in an infinitely negative cost. In contrast, the BF algorithm would detect this and terminate [192]. This algorithm can handle opposing weight edges and detect negative cycles, an advantage for specific problems. Likewise, artificial potential field (APF) is a simple technique for avoiding obstacles, but robots following this principle can get stuck in so-called local minima [37,42]. This is a time-consuming algorithm, as the robot can stop before the obstacle until it moves. The Bug algorithm is also an early version of the obstacle avoidance algorithm used in robot navigation [34]. The gap tracking method (FGM) is another early obstacle avoidance algorithm used in environments where the robot must navigate narrow spaces. Still, it can not avoid U-shaped obstacles [37,193].

Fuzzy logic (FL) has been developed among the heuristic algorithms for various applications, including obstacle avoidance robotics [71,72]. Since their initial research, genetic algorithms (GAs) have been widely applied to solve various optimization problems, including obstacle avoidance algorithms [79]. Simulated annealing (SA) and Tabu search (TS) have proven to be very effective and robust in solving a wide range of problems across various applications. They are also helpful in dealing with issues where specific parameters are not known in advance. These properties are missing in all conventional optimization techniques [88]. They apply an appropriate cost function to give feedback to the algorithm on the progress of the search. The difference in principle is how and where domain-specific knowledge is used. For example, SA obtains such information mainly from the cost function. The disturbed items are selected randomly, and the acceptance or rejection of disturbances is based on the Metropolis criterion, which is a function of cost. The cooling schedule also has a significant impact on the algorithm’s performance. It must be carefully tailored to the problem domain and the specific problem instance. TS differs from GA and SA because it has an explicit memory component. At each iteration, the neighborhood of the current solution is partially explored, and the move is made toward the best nontaboo solution in that neighborhood. The neighborhood function and the size and content of the Tabu list are problem-specific. Memory structures also influence the direction of the search. Particle swarm optimization (PSO) is a population-based heuristic optimization method derived from standing wave theory [48]. The cuckoo search algorithm (CSA) was introduced as an efficient and straightforward global search technique among evolutionary algorithms [194]. The Artificial Bee Colony (ABC) algorithm was developed to model the behavior of living organisms and is one of the evolutionary algorithms [109]. While machine learning requires human intervention, deep learning can learn from mistakes. Deep learning requires a more significant amount of data, which demands higher computational power. In deep learning, algorithms learn autonomously by analyzing large amounts of data. In contrast, reinforcement learning requires feedback from the agent to know what actions lead to the desired outcome. Significant developments in neural networks occurred in the 1980s and beyond and have been applied to obstacle avoidance robotics [125]. In control systems, the star of reinforcement learning solutions is now gone, replaced by data-driven MPC solutions that can provide theoretical guarantees of performance [195]. It is questionable whether a suitable fitness function can solve all our problems, not to mention the theoretical guarantees of stability or convergence. Nevertheless, it is worth using machine learning algorithms in engineering because, presumably, they will be able to solve more and more routine tasks for us. Furthermore, there is also the question of how data-driven MPC algorithms solve all control theory problems. Specifically, where is the space left for model-based robust control? The Hybrid Navigation Algorithm (HNA) with wandering trails combines different methods and techniques for optimal route planning, which has been applied to a partially known environment [179]. Recent developments have resulted in a New Hybrid Navigation Algorithm (NHNA) similar to the HNA. It is a complete algorithm that uses several approaches to achieve efficient and stable robot navigation. However, it cannot be used in unknown environments as it requires prior environmental information [25,193]. Sliding mode (SM) algorithms employ several methods and have seen significant development and application, especially in robotics and control systems [181].

Essential characteristics of algorithms are convergence time, computation time, and memory requirements. The convergence time is required for the algorithm to reach convergence, i.e., to achieve a stable or desired state. This time may vary depending on the algorithm type, the task’s nature, and the initial conditions. The goal is to make the algorithm converge as fast as possible to solve the task or problem efficiently. Computation time and memory requirements are closely related. The more complex the environment in which we want to navigate the robot, the more data and more complex computations are needed to find the optimum.

Among the previously developed methods, heuristic approaches are relatively new and have significant applications in mobile robot navigation. Contemporary research increasingly focuses on optimizing algorithms through hybridization to achieve superior performance. Historically, classical methodologies were prevalent but faced limitations such as susceptibility to local minima and high computational demands. In response, researchers have shifted towards heuristic methods, particularly effective in uncertain or unknown environments. These heuristic approaches, often enhanced by hybridization with classical methods, have proven successful in complex three-dimensional workspaces, such as those encountered by underwater, unmanned aerial vehicles, and humanoid robots. This shift underscores the improved adaptability and efficiency of heuristic strategies over classical approaches in dynamic settings.

As technology advances and robotics becomes increasingly integrated into various aspects of our lives, obstacle avoidance algorithms are poised to undergo significant developments to meet the demands of emerging applications. With the proliferation of machine learning techniques, we can expect obstacle avoidance algorithms to incorporate more advanced learning-based approaches. These algorithms will be capable of adapting and improving their performance over time through experience and feedback, leading to more efficient and robust obstacle avoidance in dynamic environments. Future obstacle avoidance systems will rely on sophisticated sensor fusion techniques to integrate data from multiple sensors, such as LIDAR, cameras, radar, and ultrasonic sensors. By combining information from diverse sources, these algorithms will achieve a more comprehensive understanding of the environment, enhancing their ability to detect and avoid obstacles accurately. Future obstacle avoidance algorithms prioritize real-time adaptive planning to navigate complex and dynamic environments effectively. These algorithms will continuously analyze sensor data and adjust robot trajectories to avoid obstacles and navigate changing scenarios in real-time, unreal-time, and efficient robot operation. Collaborative obstacle avoidance algorithms will become increasingly important in environments where multiple robots or autonomous vehicles operate concurrently. These algorithms will enable robots to communicate and coordinate their movements to avoid collisions and optimize path planning, leading to smoother and more efficient operations in shared spaces. Drawing inspiration from nature, future obstacle avoidance algorithms may incorporate bio-inspired principles, such as swarm intelligence or mimicry of animal behavior. These approaches could lead to innovative solutions for navigating challenging environments, leveraging the collective intelligence of swarms, or mimicking the agility and adaptability of animals in natural habitats. As robots become more prevalent, ethical considerations regarding obstacle avoidance will gain prominence. Future algorithms must balance efficiency with moral considerations, prioritizing human safety and well-being in crowded environments. Additionally, human-robot interaction will play a crucial role, with obstacle avoidance algorithms designed to anticipate and respond effectively to human intentions and behaviors. With these exciting developments, researchers and engineers can pave the way for safer, more efficient, and more adaptive robotic systems in various applications.

## Figures and Tables

**Figure 1 sensors-24-03573-f001:**
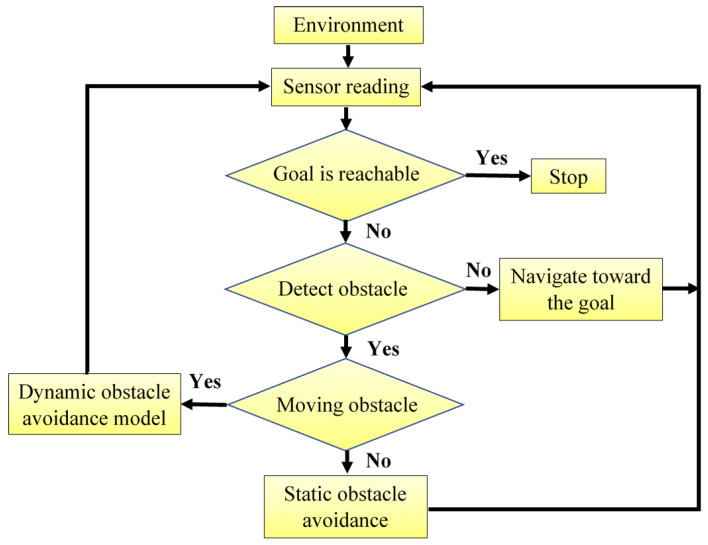
The obstacle avoidance procedure [4].

**Figure 2 sensors-24-03573-f002:**
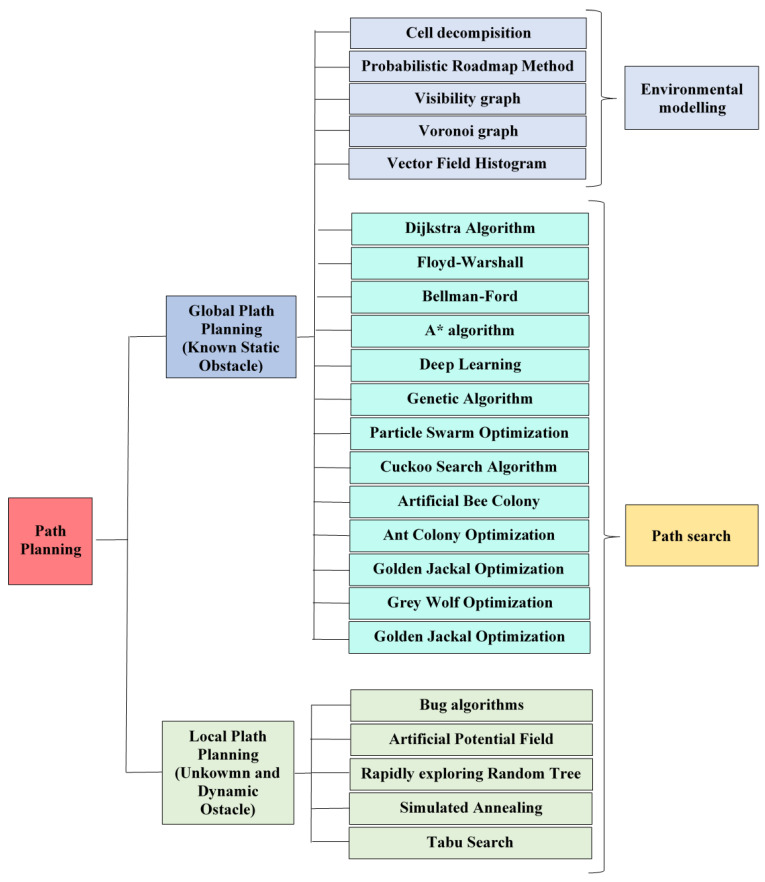
Diagram of the algorithms.

**Figure 3 sensors-24-03573-f003:**
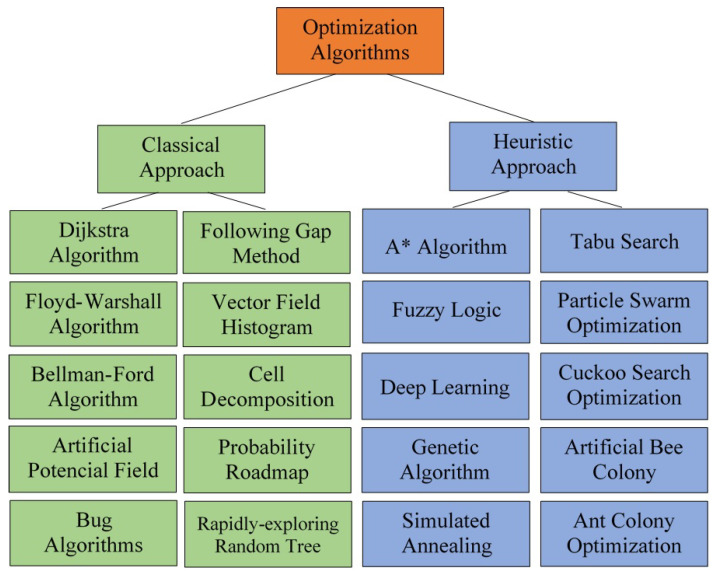
The classical/heuristic division of algorithms.

**Figure 4 sensors-24-03573-f004:**
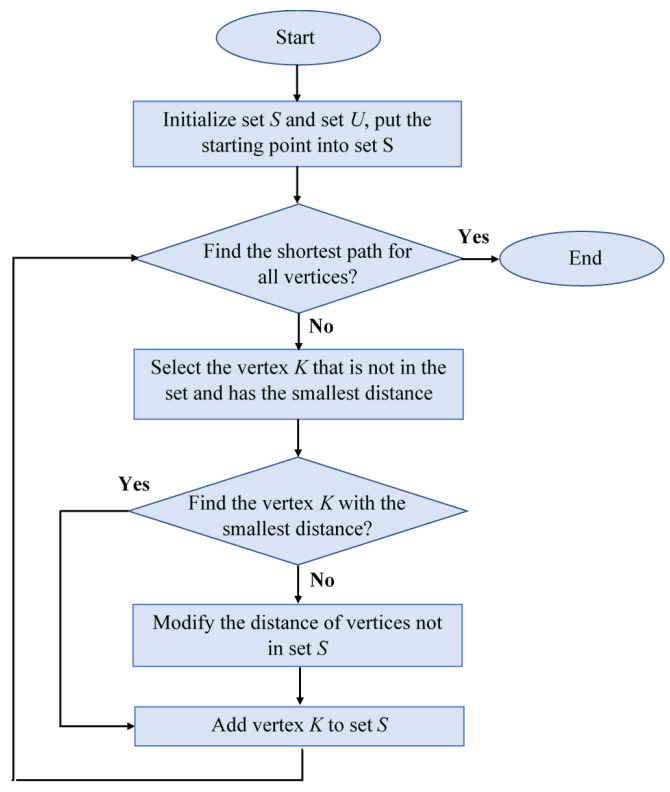
The flowchart of the Dijkstra.

**Figure 5 sensors-24-03573-f005:**
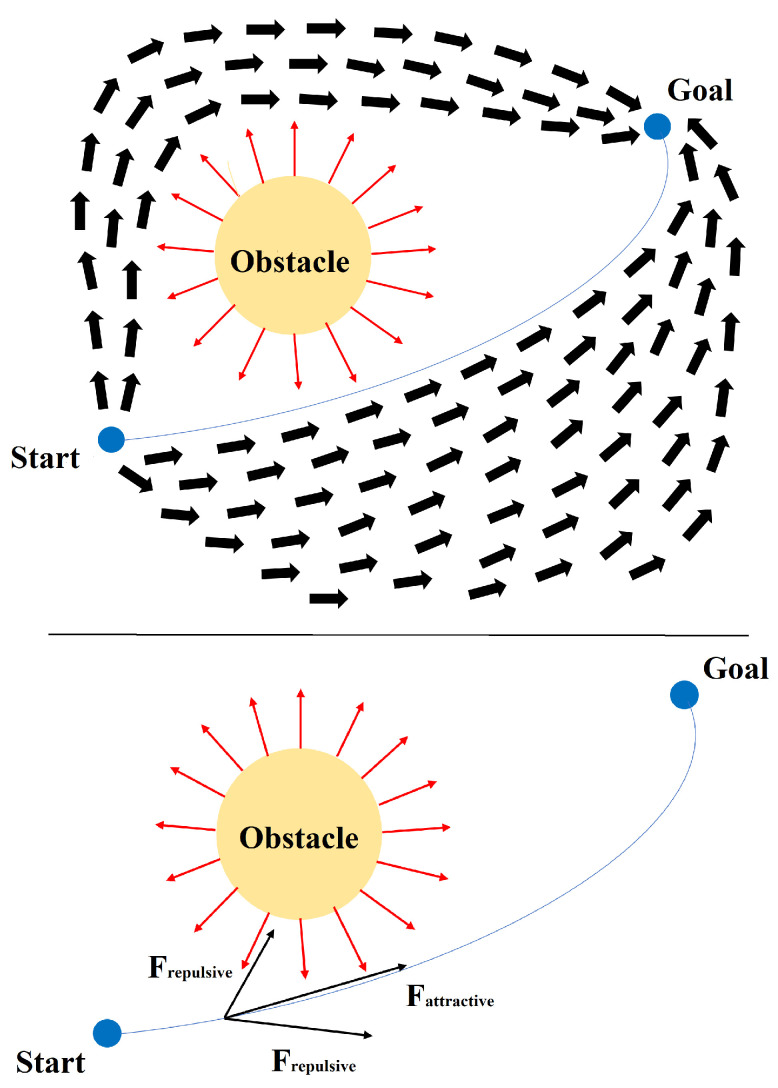
APF-based navigation for a mobile robot.

**Figure 6 sensors-24-03573-f006:**
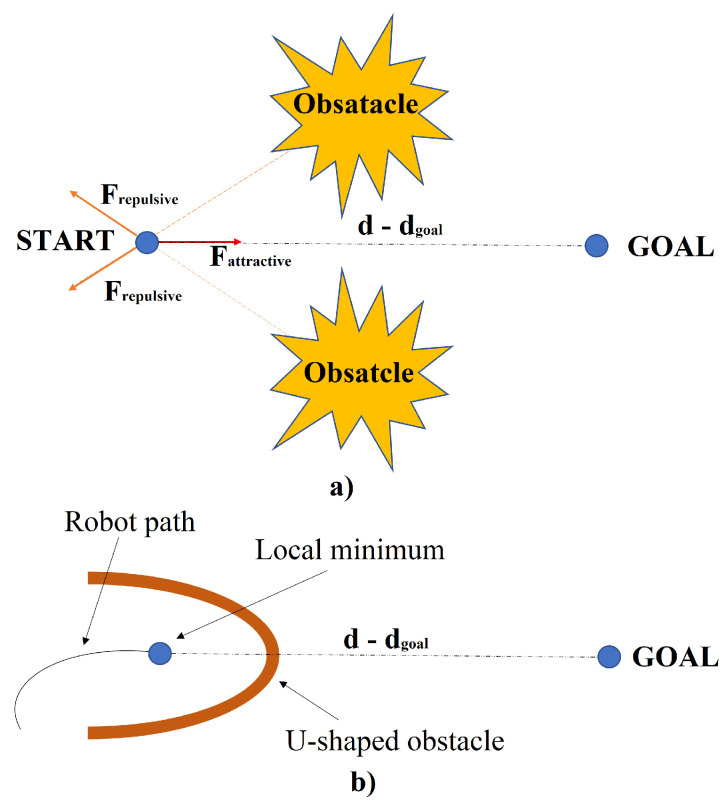
Dead-end scenario of the artificial potential field method: (**a**) symmetric obstacles and (**b**) U-shaped obstacle.

**Figure 7 sensors-24-03573-f007:**
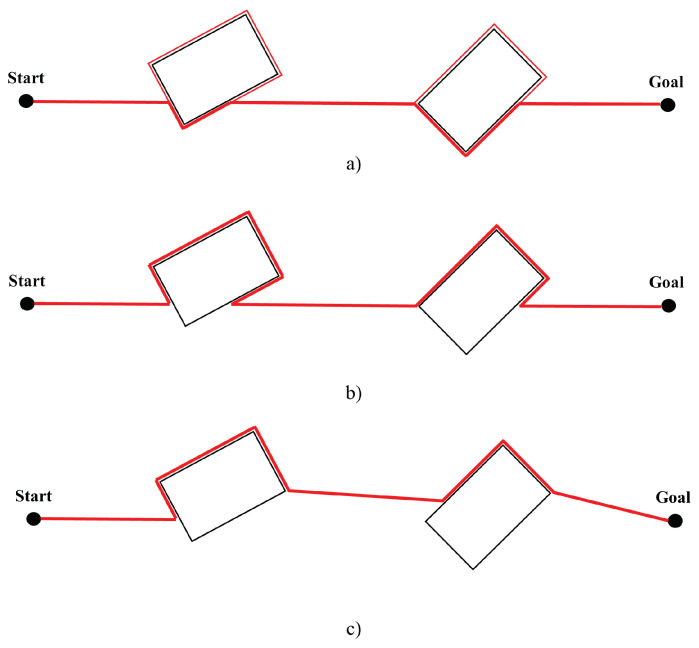
Obstacle avoidance with the Bug algorithms: (**a**) path of the Bug-1 algorithm, (**b**) the path of the Bug-1 algorithm, and (**c**) Dist-Bug algorithm path.

**Figure 8 sensors-24-03573-f008:**
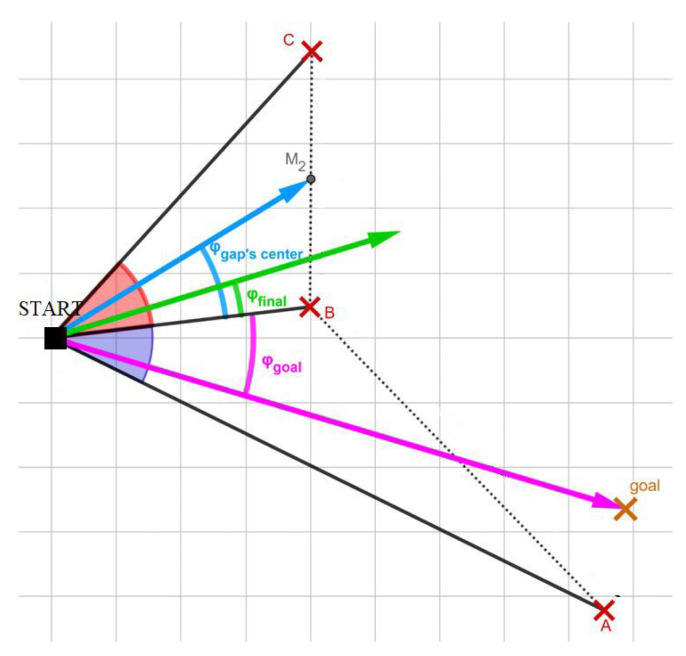
Robot-obstacle configuration, obstacles (*A*, *B*, and *C*), the midpoint of the widest angular gap ($M_{2}$), goal point (*X*), angle to the goal point ($\phi_{goal}$), final heading angle ($\phi_{final}$), and angle to the largest gap’s center point ($\phi_{gap^{\prime }scenter}$). [38].

**Figure 9 sensors-24-03573-f009:**
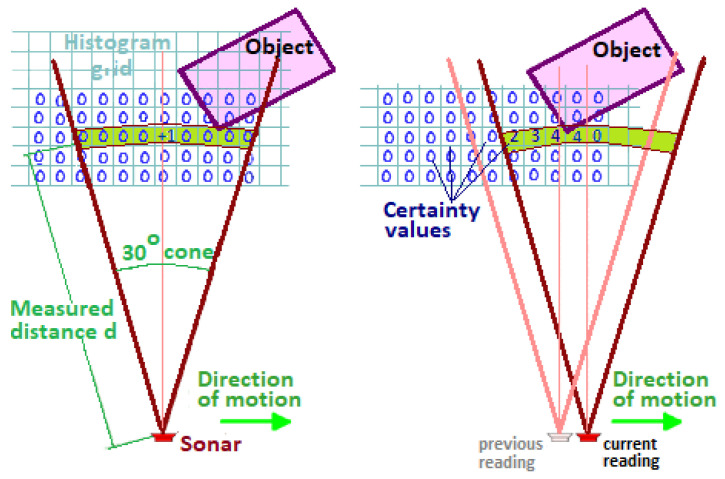
Structure of the 2D histogram grid map [42].

**Figure 10 sensors-24-03573-f010:**
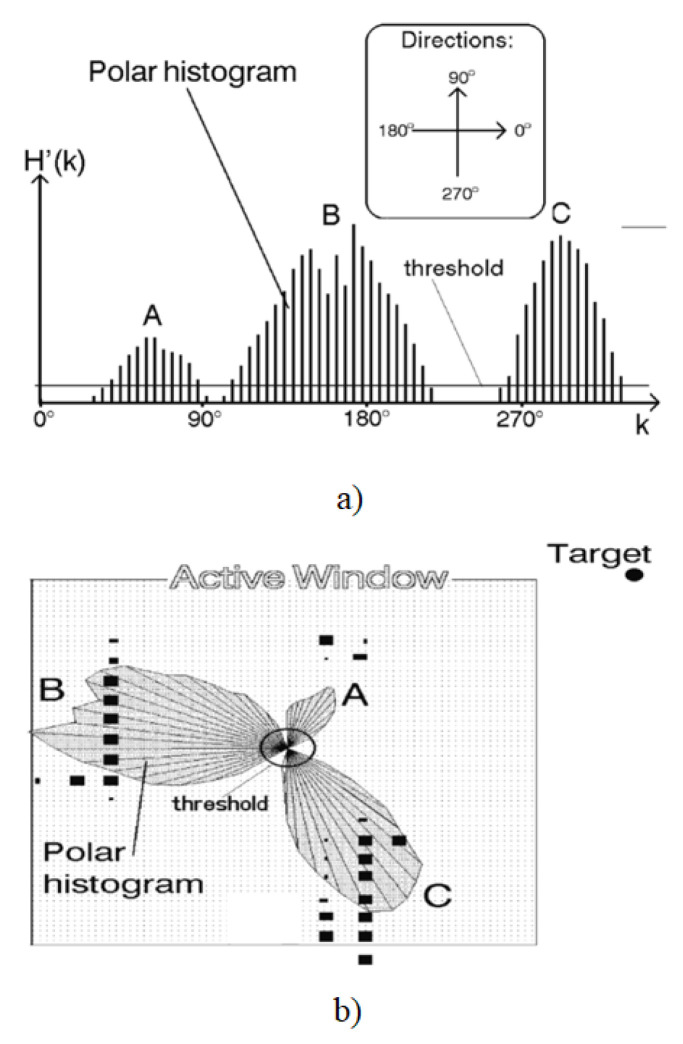
Representation of (**a**) the 1D histogram (**b**) the polar histogram with obstacle density for a situation where the robot has three obstacles, *A*, *B* and *C* in its close vicinity [42].

**Figure 11 sensors-24-03573-f011:**
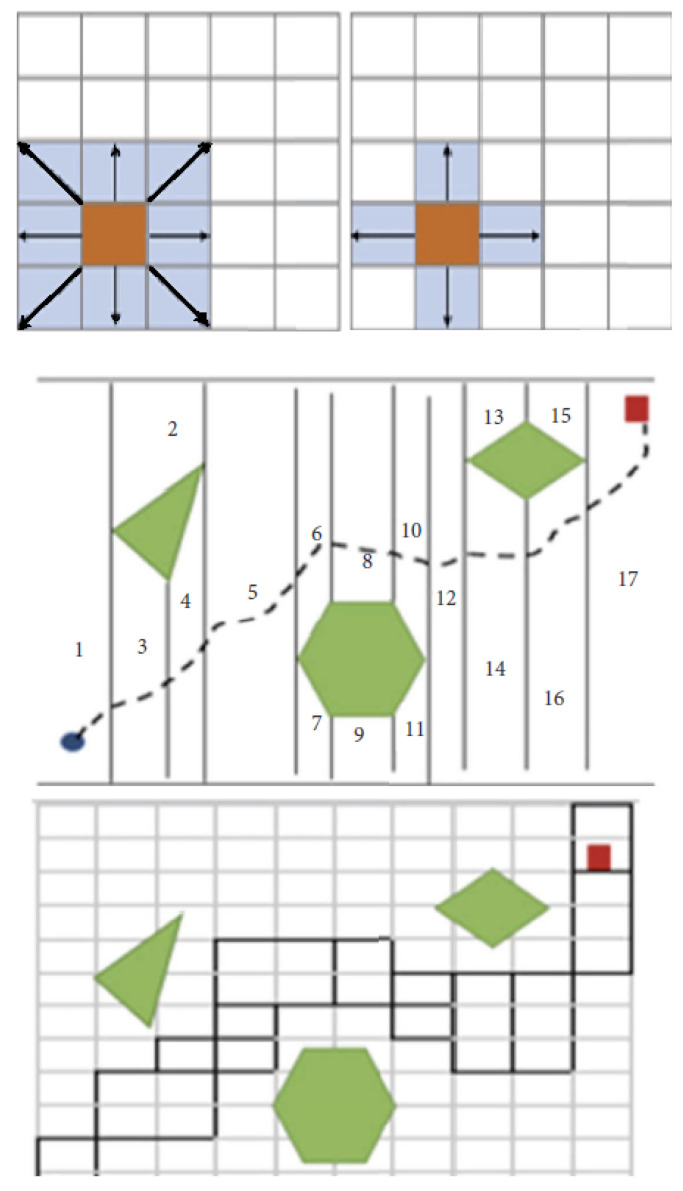
(**Top**): approximate CD; (**middle**): exact CD, and (**bottom**): probability CD [47].

**Figure 12 sensors-24-03573-f012:**
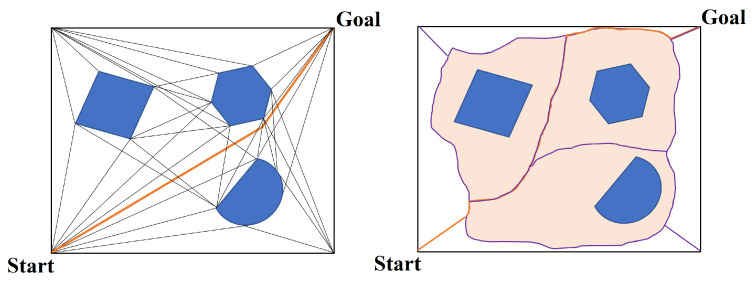
Visibility graph (**left**); Voronoi graph (**right**). The visibility graph is constructed based on the visibility between points, while the Voronoi graph is constructed based on the geometric relationships between areas.

**Figure 13 sensors-24-03573-f013:**
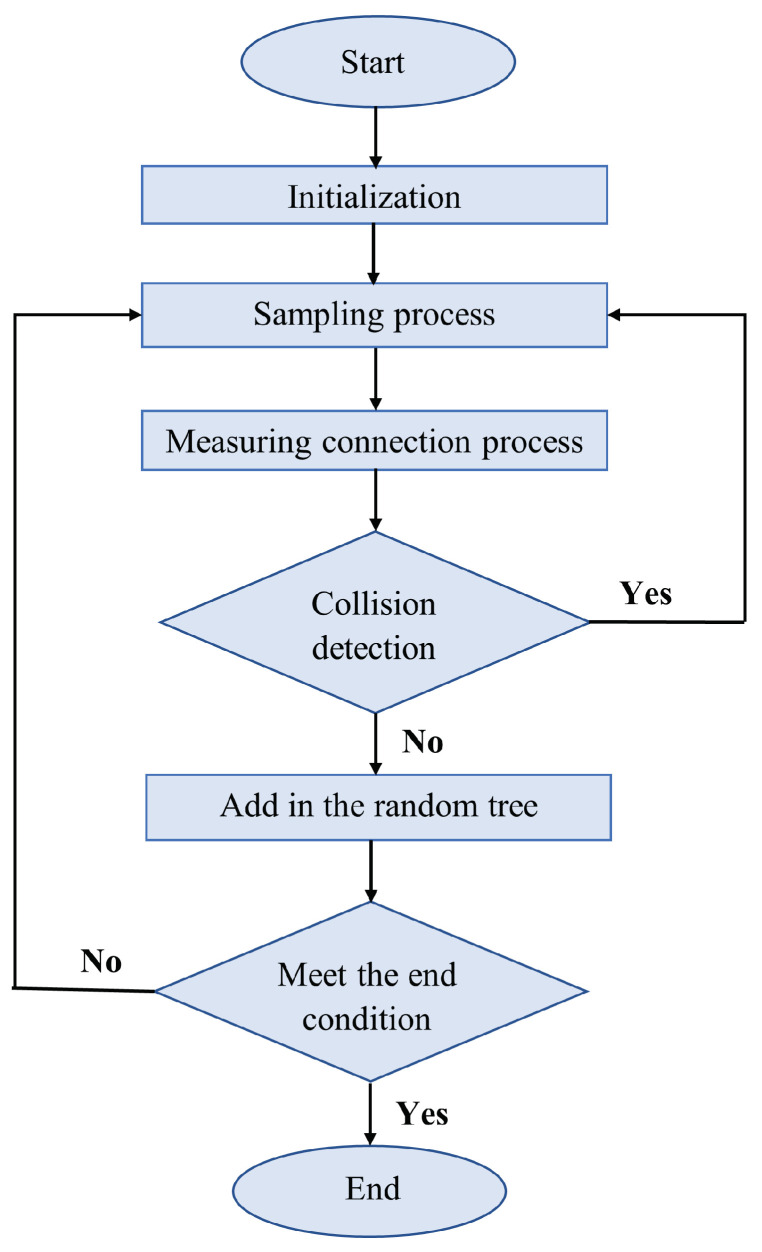
The RRT process.

**Figure 14 sensors-24-03573-f014:**
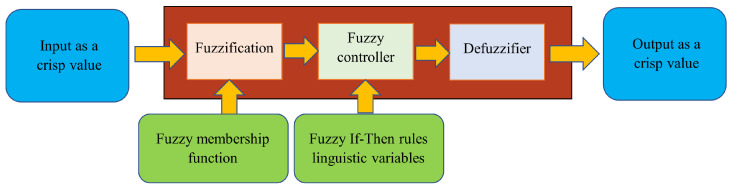
Basic FL controller consisting of an IF–THEN rule, an inference mechanism, an input fuzzification unit, and an output defuzzification unit.

**Figure 15 sensors-24-03573-f015:**
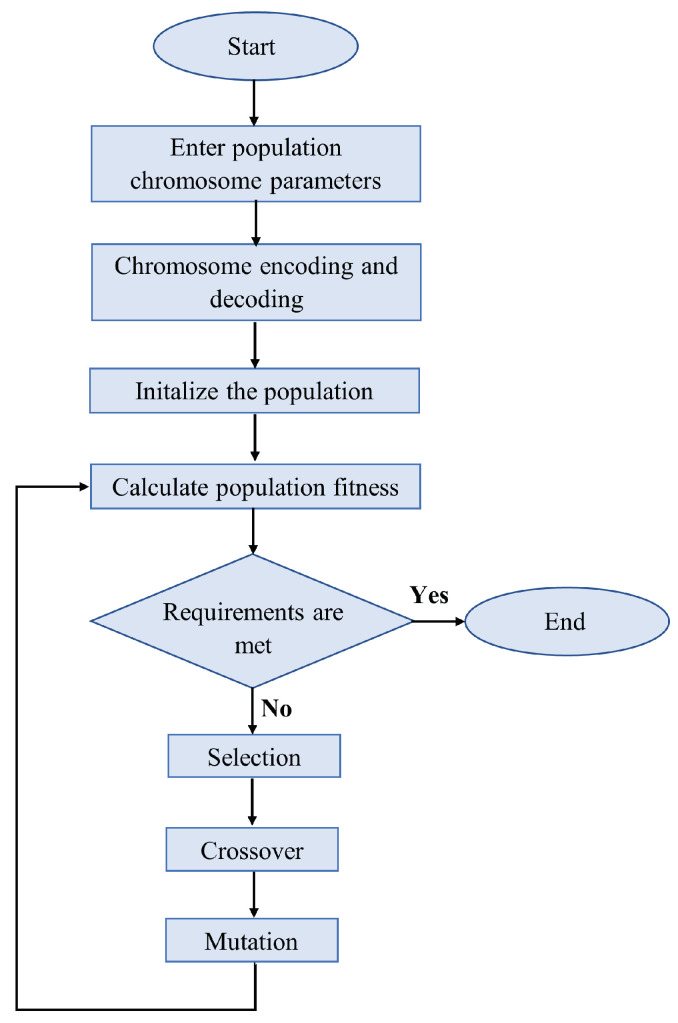
Process of GA [80].

**Figure 16 sensors-24-03573-f016:**
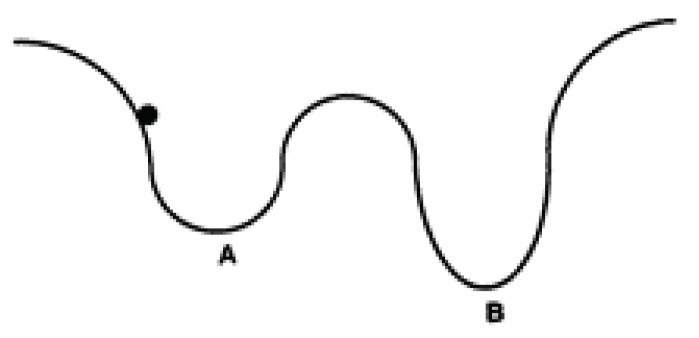
The potential barriers (A, B).

**Figure 17 sensors-24-03573-f017:**
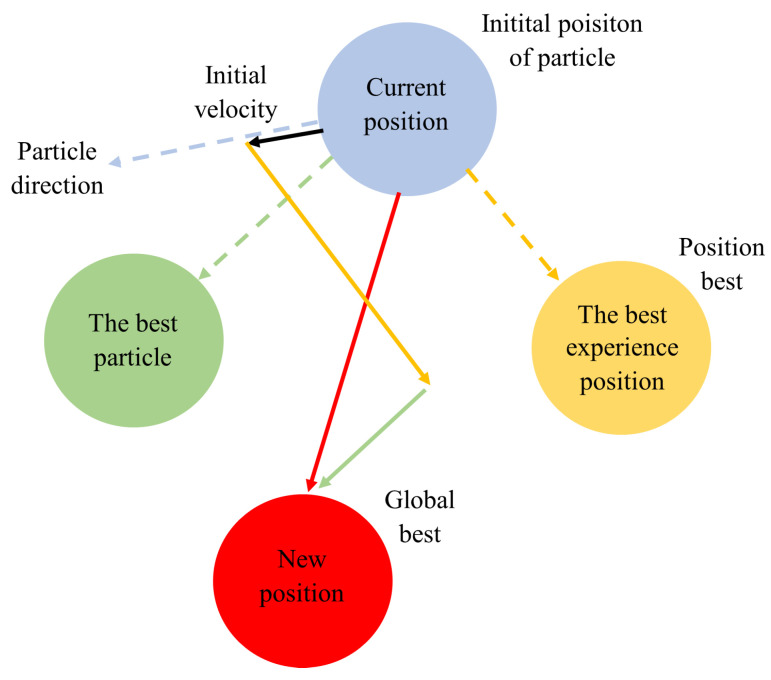
The basic concept of PSO.

**Figure 18 sensors-24-03573-f018:**
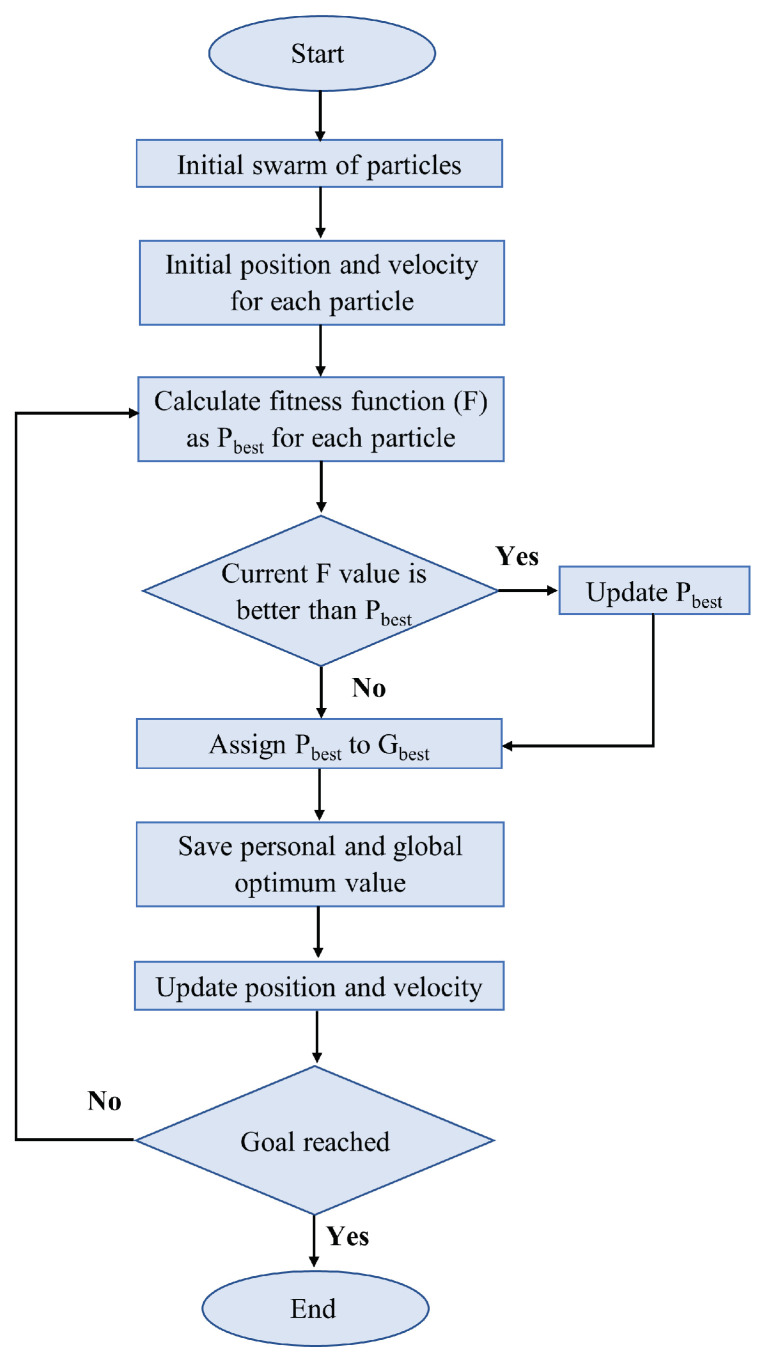
The PSO process.

**Figure 19 sensors-24-03573-f019:**
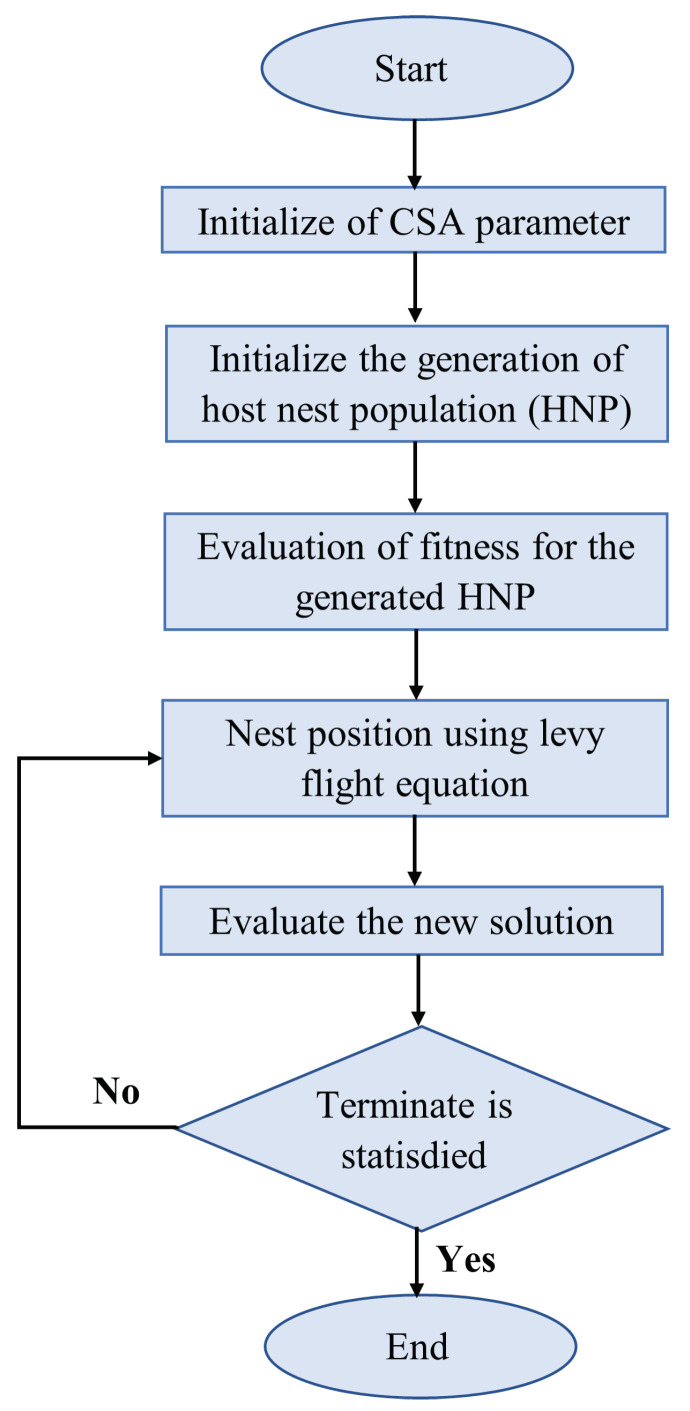
The CSA process.

**Figure 20 sensors-24-03573-f020:**
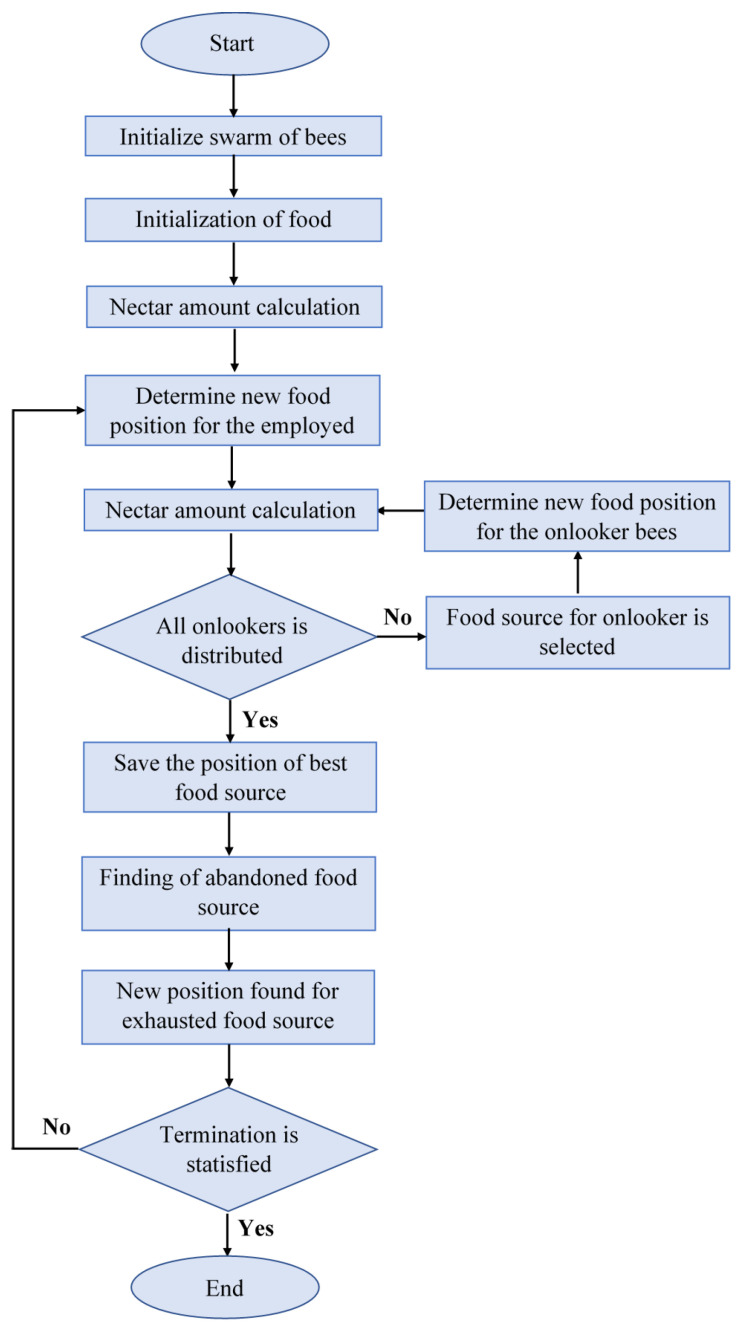
The ABC process.

**Figure 21 sensors-24-03573-f021:**
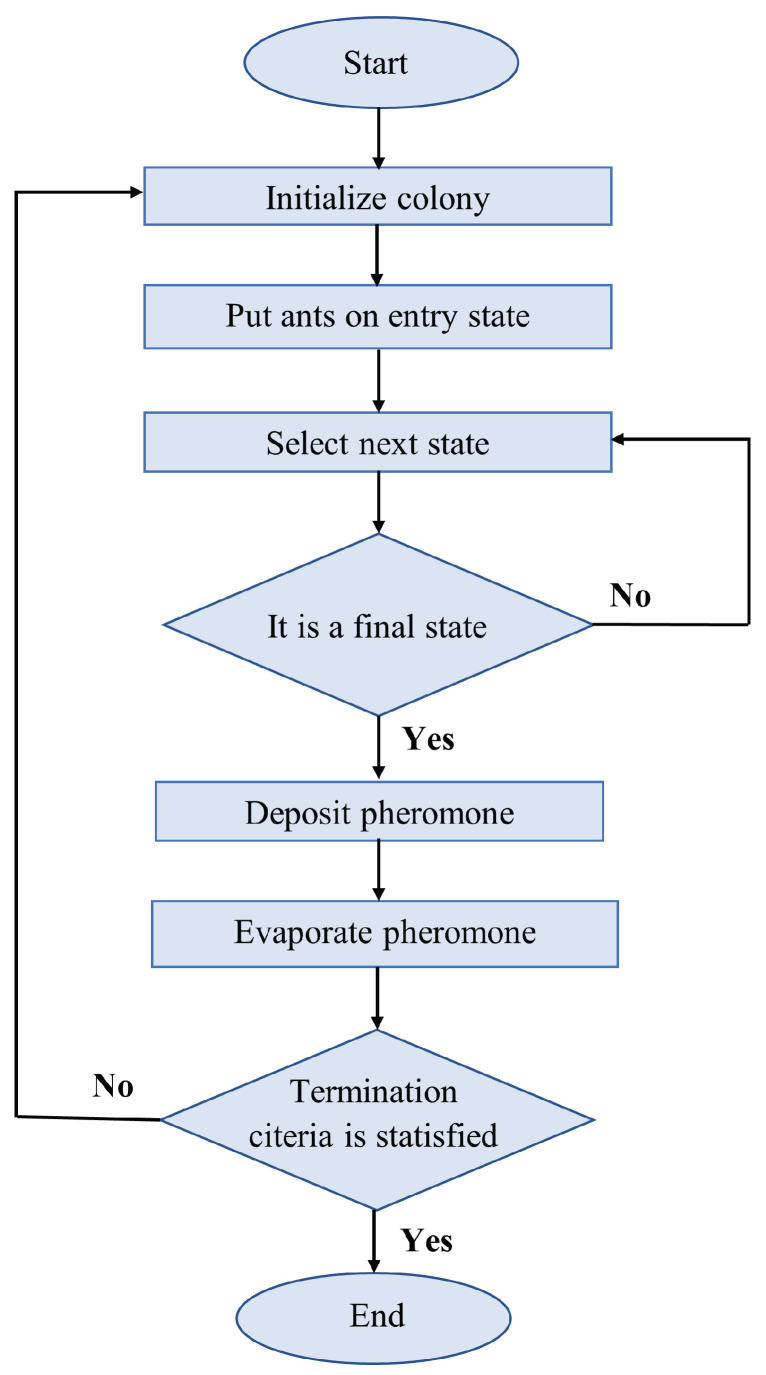
The ACO process.

**Figure 22 sensors-24-03573-f022:**
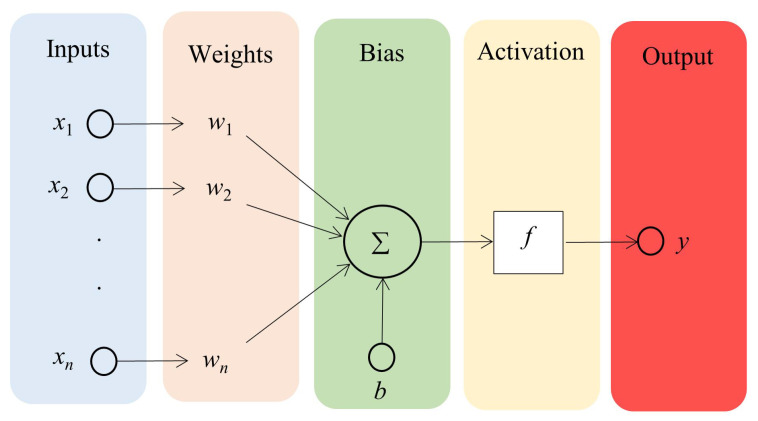
A general neuron structure.

**Figure 23 sensors-24-03573-f023:**
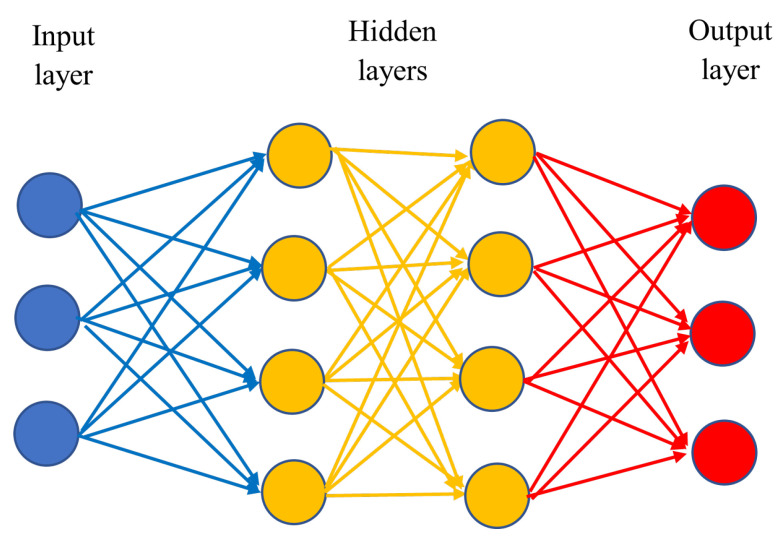
A possible structure of the ANN (the number of hidden layers may vary).

**Figure 24 sensors-24-03573-f024:**
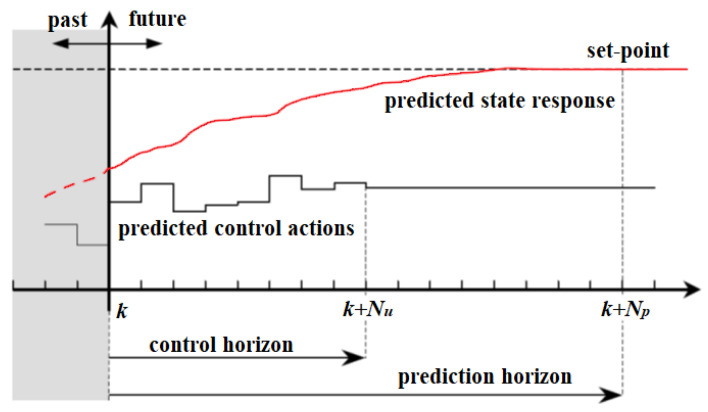
The general concept of the MPC.

**Figure 25 sensors-24-03573-f025:**
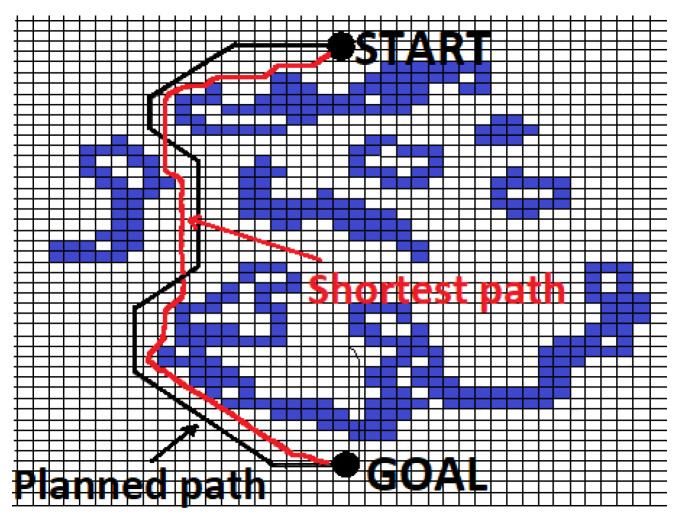
Raster map and robot path (with NHNA) [25].

**Figure 26 sensors-24-03573-f026:**
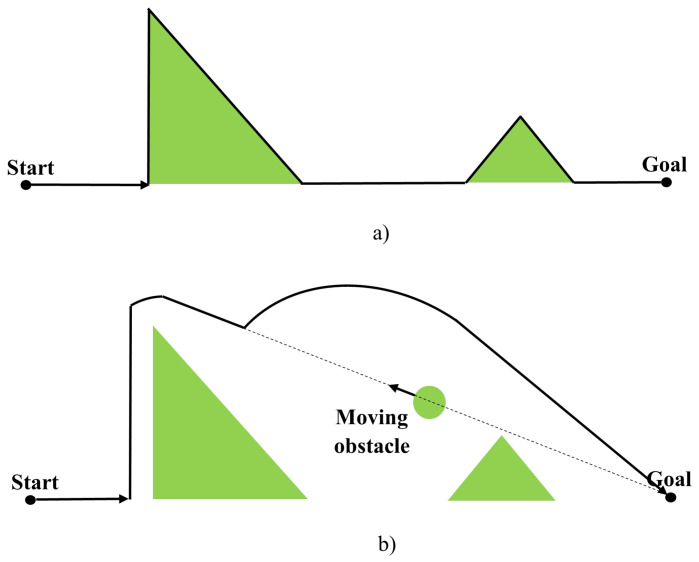
Obstacle avoidance strategy: (**a**) path of the Dist-Bug algorithm and (**b**) robot trajectory with the distance histogram (D-H) error algorithm.

**Figure 27 sensors-24-03573-f027:**
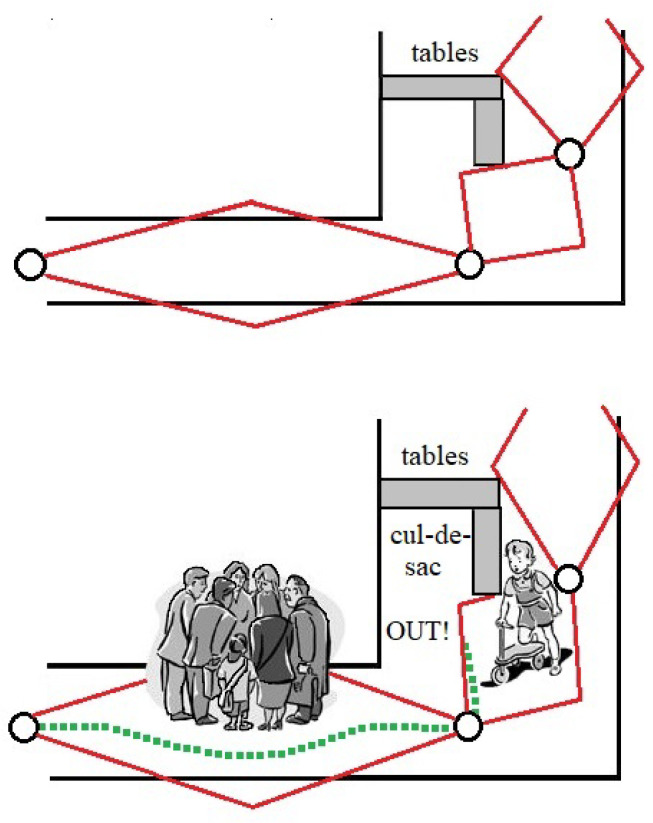
Route Path of the robot with roaming trails (HNA): Top: Primary map with roaming trails. Bottom: Robot trajectory (dashed line) [179].

**Figure 28 sensors-24-03573-f028:**
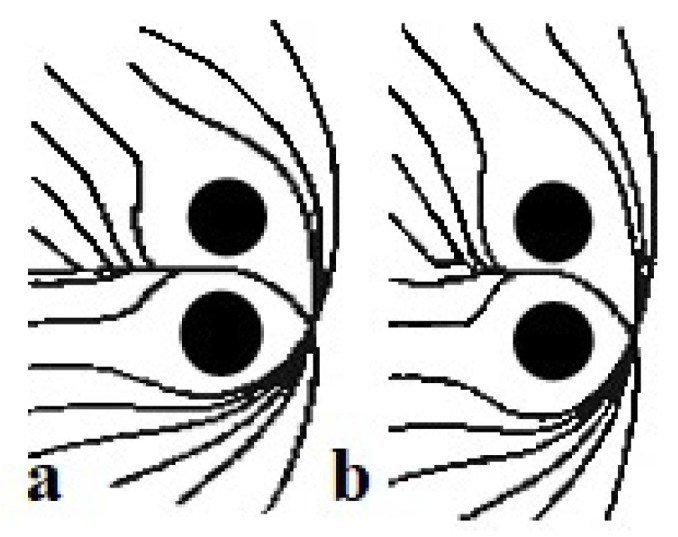
Gradient lines for switching noncontinuous gradients: (**a**) noncontinuous switching; (**b**) smooth switching [24].

**Table 1 sensors-24-03573-t001:** Summary table of the classical algorithms in the thesis. Part 1.

Algorithm	Advantages	Disadvantages	Convergence Time	Calculation Time	References
Dijkstra	Robust and reliable operation; Ensuring accurate route planning	Not adaptable to dynamically changing environments	Long	High	[5]
FW	Find the shortest route between all pair nodes	High memory requirements for large graphs; $O(n^{3})$ running time, which is inefficient for large graphs	Medium	High	[19]
BF	Ability to handle negative weights; Detects negative cycles	Slower than Dijkstra for positive weight graphs; $O(n\dot{m})$ running time	Medium	Medium [20,21,22]	
APF	Simple and intuitive method; Ability to handle both static and dynamic obstacles	The robot can get stuck in local minima; Difficult to use in more complex environments	Medium	High	[23]
Bug	Simple and easy to implement algorithm; Good for avoiding static obstacles	No guarantee of the shortest route; Less effective for more complex or dynamic obstacles	Short	Low	[33,34]

**Table 2 sensors-24-03573-t002:** Summary table of the classical algorithms in the thesis. Part 2.

Algorithm	Advantages	Disadvantages	Convergence Time	Calculation Time	References
FGM	Very effective on narrow or fragmented gaps	Not effective on every obstacle; Difficult to use in confined maps or with large robots	Short	Low	[37,38]
VFH	Flexibility and adaptability;	Time- and computation-intensive, especially for large maps; Complex parameterization	Medium	High	[41,43]
CD	It effectively bypasses local minima to help find globally optimal solutions	High computational demand and memory requirements; Proper parameterization and fine-tuning can be critical for efficiency	High	High	[46]
PRM	Integrate sensory data and probabilistic information	High computational and memory demand; Complex parameterization and fine-tuning	Medium	High	[48,52]
RRT	Suitable for solving the route planning problem in dynamic and multi obstacle conditions; applicable to the route planning problem in high-dimensional environments	The route is randomly generated, the route is biased; The convergence speed is slow, and the search efficiency is low	High	High	[56,58,60]

**Table 3 sensors-24-03573-t003:** Summary table of the heuristic algorithms in the thesis. Part 1.

Algorithm	Advantages	Disadvantages	Convergence Time	Calculation Time	References
A*	Direct search; No preprocessing required	Large amount of calculation; Optimal solution not guaranteed	Medium	Medium	[63]
FL	A flexible and adaptable; React to uncertainties and foggy information	High memory requirements; the rules and parameters largely require human intervention	Medium	High	[69,75]
GA	Strong global searching ability	Slow convergence; Poor local optimization; Poor stability	Long	High	[50,79]
SA	Good for global optimization; ability to avoid local minima	Global optimum is not guaranteed; Depends on cooling schedule	Medium-Long	Medium	[88,90]
TS	Ability to avoid local minima; can be used for complex problems	High memory requirements due to the taboo list; parameter-sensitive	Medium	Medium-High	[88]
PSO	Fast search time; high convergence speed in early-stage	Slow convergence speed in later period; easy to fall into local optimum	Medium	High	[92,93,94]
CSA	Simple and easy to implement; efficient exploration and optimization of space	No guarantee of a global optimal solution; Less effective for more complex or large problems	Medium	Medium	[102,103,104]
ABC	Flexible and adaptable; finding global optimal solutions for larger systems;	Parameterization and fine-tuning is time-consuming; High memory requirements	High	High	[109,110,112]
ACO	Strong global searching ability; high efficiency; high convergence speed in later period	Slow convergence speed in early stage	High	High	[80,120]

**Table 4 sensors-24-03573-t004:** Summary table of the heuristic algorithms in the thesis. Part 2.

Algorithm	Advantages	Disadvantages	Convergence Time	Calculation Time	References
DWA	Fast response times in real-time applications; efficient local obstacle avoidance	Finding only local solutions; global optimum is not guaranteed	Low	Low	[165,166,167]
GJO	Powerful global search capability; ability to avoid local minima	Requires significant computing resources; sometimes slower convergence	Medium-Long	High	[169,170]
GWO	Powerful global search capability; handles multidimensional optimization problems well	Possible early convergence; depends on fine-tuning of parameters	Long	Medium	[172,173]
GSA	Good global search capability; robust for different types of problems	Slow convergence; requires significant computing resources	Long	High	[177,178]
ANN	Ability to learn and adapt; robust and able to handle large amounts of data	High memory requirements; during the learning phase, large data sets are needed	Long	High	[125,126]
MPC	Forward-looking optimization; ability to manage the limitations of systems	High computing resources; complex implementation	Medium-Long	High	[154]
DRL	Complex problem solving; autonomous learning; good generalization ability	High computational demand; high data demand	Long	High	[124]

**Table 5 sensors-24-03573-t005:** Summary table of the hybrid algorithms in the thesis.

Algorithm	Advantages	Disadvantages	Convergence Time	Calculation Time	References
NHNA	Integrate the benefits of multiple algorithms; improved accuracy and efficiency in different environments	More complex implementation; high calculation demand	Medium	Medium-High	[25]
HNA	Better route optimization; more flexibility in dealing with obstacles	Requires significant computing resources; complex parameter tuning	Medium	Medium-High	[44,179]
SM	Robust in unknown dynamic environments; fast reaction time	Sensitive to noise and discontinuities; Precise modeling required	Short-Medium	Medium	[181]

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
