# Peer review of "Obstacle Avoidance and Path Planning Methods for Autonomous Navigation of Mobile Robot"

_sensors, 2024, doi:10.3390/s24113573_

Round 1
Reviewer 1 Report
Comments and Suggestions for Authors
The paper presents a comprehensive analysis of emerging trends in obstacle avoidance algorithms, encompassing various dimensions including technological advancement, potential applications, and ethical considerations. The paper exhibits a clear logical structure, yet the review process has identified areas necessitating refinement to enhance overall quality.
1. Within the "Classic approaches" segment, some algorithms such as Floyd-Warshall and Bellman-Ford, and within the "Heuristic approaches" category, algorithms such as Simulated Annealing and Tabu Search, alongside Hybrid algorithms featuring Deep Reinforcement Learning for Hybrid Routing, showcase distinctive merits and application domains. It is advisable to provide succinct elucidations on these algorithms to enrich the discussion.
2.Tables 1 and 2 serve as concise platforms for delineating the attributes of classical and heuristic algorithms, respectively. This methodological presentation proves effective. It is recommended to employ a similar strategy to briefly encapsulate the traits of Hybrid algorithms, thus enhancing readability.
3.The flowchart depicted in Figure 4 contains a logical flaw, as it comprises solely one termination block. Rectification of this issue is warranted. A parallel issue is evident in Figure 21 and warrants correction.
These adjustments will contribute to the refinement and coherence of the paper, aligning it more closely with academic standards.
Comments on the Quality of English LanguageThe overall language expression is clear and fluent, but some sentence structures are slightly complex. It is recommended to simplify the language expression to improve the reader's understanding.
Author Response
Thank you for the questions formulated in the critique, which contribute to raising the quality of the thesis and the research potential inherent in them.
- Algorithms that prove to be useful in practice were not mentioned in the thesis within the above-mentioned segments. We have revised the mentioned sections.
- The paper does not contain a summary table of the characteristics of the hybrid algorithms discussed in more detail. In the paper, we created a new table for determining the characteristics of hybrid algorithms.
- The flowchart shown in Figure 4 and Figure 21 is incorrect. We have updated the figures.

Reviewer 2 Report
Comments and Suggestions for Authors
The article provides an overview of key obstacle avoidance algorithms. The advantages, limitations and application areas of these obstacle avoidance algorithms are presented. However, there are some problems with the paper.
1.There are many reviews on robot path planning and obstacle avoidance. Please explain the biggest differences and advantages between this review and other similar reviews.
2.The relevant work analysis is not in-depth enough. For example, in the third paragraph of Page 2 there is a statement "This paper follows the latter classification". Please add some material to explain why this classification is used and explain the advantages of this classification.
3. The second and third paragraphs of Page 2 contain only one sentence, which is not very typical of a scientific paper. Please expand or combine it into another paragraph. There are many similar problems in the article, please check the full text for correction.
4.There are many format errors in the paper. For example, the first paragraph of Page 2 has the statement "ontinuous knowledge of the target's position relative to the robot is critical for accurate navigation, as depicted in Figure 1 ". What does the word "ontinuous" mean ? The word is also not capitalized. Please check the full text carefully and correct similar errors.
5.Part information in Figure 2 is incomplete, such as "(Unknown and)". Part information is wrong, such as "Local plath Planning".
6.The composition of the paper is not reasonable, some pictures occupy the whole page.
7. The author seems to lack experience in writing scientific papers. Some parts of the language in the article are not very smooth. It is recommended to have a professional review and polish the paper.
Comments on the Quality of English LanguageExtensive editing of English language required
Author Response
Thank you for the questions formulated in the critique, which help to clarify the advantages of the paper, and contribute to raising the quality of the paper, and thank you for the research opportunity inherent in them.
- This paper attempts to provide the most comprehensive overview of all algorithms that are historically important and currently significant in practice. Of course, there is no chance to discuss all possible methods (especially in the case of hybrid algorithms), but we tried to present the development directions for each algorithm. Such a large amount of literature reviews cannot be found in other journals. One of the greatest values ​​of the article is the summary table of the individual algorithms after the theoretical background, which provides a sufficient comparison based on the main properties characteristic of the algorithms (e.g. convergence, calculation time)). We have updated the manuscript by rewording the introduction.
- The classification according to classic and heuristic obstacle avoidance algorithms makes the selection and application of algorithms more transparent and manageable. Users can more easily identify which algorithm best meets the requirements of a given problem. Classical algorithms such as Dijkstra perform well for small deterministic problems, while heuristic algorithms such as A* can be more efficient for larger and more complex problems. Moreover, the separation according to global and local search algorithms is less clear. There are heuristic algorithms (such as A* or the DL-based algorithms) that have both versions of the search algorithm. We have updated the manuscript by rewording the introduction.
- In the thesis, we found several one- or two-sentence paragraphs, which impairs the structural quality and coherence. We thoroughly reviewed the entire text to identify and correct other instances of single-sentence paragraphs, ensuring proper structure and coherence throughout the paper.
- During the review of the paper we discovered several formatting errors that impair the comprehensibility of the text. We carefully reviewed the entire manuscript for formatting errors and corrected them. We changed the word "ontinuos" to "Continuos".
- Figure 2 is not complete and needs to be supplemented. We have updated the figure.
- While reviewing the thesis, we found some large images that took up the entire page, which spoiled the structure of the paper. We have updated the figures.
- Some parts of the language in the article are not very smooth. We reviewed the thesis professionally, in those cases where it was difficult to understand the text, we revised the text.

Reviewer 3 Report
Comments and Suggestions for Authors
The paper makes a detailed survey of the existing obstacle avoidance and path planning methods in the field of robotics. However, after careful evaluation, I regret to inform you that the paper cannot be published in its current form for several reasons:
1. Only these basic artificial intelligent optimization algorithms are covered, but these modified versions also need to be included and analyzed.
2. It seems that the block of local path planning in Fig.2 is not complete.
3. For recent research trends, deep reinforcement learning methods are useful tools for collision avoidance and path planning tasks, but the authors do not carry out any corresponding survey.
4. Some figures are blurred and they should be redrawn, such as Fig. 25 and Fig. 26.
5. Grammatical errors are present, and attention to English tense consistency is necessary for the author to address.
Comments on the Quality of English LanguageGrammatical errors are present, and attention to English tense consistency is necessary for the author to address.
Author Response
Thank you for the questions formulated in the critique, which contribute to raising the quality of the paper and the research potential inherent in them.
- This review was prepared with a foundational intention. We think that it is impossible to analyze each improved algorithm in more detail, so here we have only highlighted the development directions of individual methods and their possible joint application with other algorithms. Several of the optimization procedures based on artificial intelligence were discussed in detail, but some of them were not included in the summary table. We expanded the summary tables by analyzing additional algorithms that are included in more detail in the paper. Also, the paper was supplemented with a further analysis of deep learning.
- Figure 2 is not complete and needs to be supplemented. We have updated the figure.
- Deep reinforcement learning was tangentially mentioned in the artificial neural networks subsection, however, due to its spread and practical use nowadays, it is worth giving a longer analysis.A separate subsection is devoted to the analysis of the deep reinforcement learning method.
- Figures 25 and 26 are a bit blurry. We have updated the figures.
- After carefully reviewing the paper, we were able to identify several grammatical errors. We have completed the grammatical correction of the paper.

Round 2
Reviewer 1 Report
Comments and Suggestions for Authors
The author has basically solved my question, and now I have no other questions
Comments on the Quality of English LanguageMinor editing of English language required
Reviewer 3 Report
Comments and Suggestions for Authors
No more comments.